

# A new and very spiny lizard (Gymnophthalmidae: *Echinosaura*) from the Andes in northwestern Ecuador

Mario H. Yánez-Muñoz[1,2], Omar Torres-Carvajal[3], Juan P. Reyes-Puig[1,2], Miguel A. Urgiles-Merchán[1] and Claudia Koch[4]

[1] Unidad de Investigación, Instituto Nacional de Biodiversidad (INABIO), Quito, Ecuador
[2] Fundación Ecominga Red de Protección de Bosques Amenazados, Fundación Oscar Efrén Reyes, Departamento de Ambiente, Baños, Ecuador
[3] Museo de Zoología, Escuela de Ciencias Biológicas, Pontificia Universidad Católica del Ecuador, Quito, Ecuador
[4] Herpetology, Zoologisches Forschungsmuseum Alexander Koenig (ZFMK), Bonn, Germany

## ABSTRACT

We describe a new species of Neotropical spiny-lizard of the genus *Echinosaura* from the Imbabura and Carchi Provinces on the western slopes of the Andes in northwestern Ecuador. The new species mostly resembles *E. horrida*. However, it can be distinguished from all congeners by having keeled enlarged dorsal scales forming a paired vertebral row, two paravertebral series of short oblique rows of projecting scales, and a pair of spine-like scales on temporal and nuchal regions. We also provide a detailed description of the osteology of the skull and pectoral girdle of the new species and present a phylogenetic hypothesis for *Echinosaura* based on three mitochondrial genes (12S, 16S, ND4) and one nuclear gene (c-mos).

Corresponding author
Claudia Koch, c.koch@zfmk.de, c.koch@leibniz-zfmk.de

## INTRODUCTION

Neotropical spiny-lizards traditionally ranked as the genus *Echinosaura* Boulenger 1890 are small riparian lizards occuring west of the Andes from Panama to Ecuador (*Vásquez-Restrepo et al., 2020*). Most species of *Echinosaura* have partially compressed tails and bear enlarged tubercles and strongly keeled scales on body and tail, often forming crests, which represent adaptations for aquatic locomotion (*Marques-Souza et al., 2018*).

Recent molecular phylogenetic studies have resulted in major taxonomic changes within *Echinosaura*, such as synonymy of *Teuchocercus* Fritts & Smith 1969 with *Echinosaura* and description of *Centrosaura* Uzzell 1966 and Rheosaurus Vásquez-Restrepo, Ibáñez, Sánchez-Pacheco & Daza 2020, to accommodate "*E. apodema*" Uzzell 1966 and "*E. sulcarostrum*" Donnelly, Macculloch, Ugarte & Kizirian 2006, respectively (*Torres-Carvajal et al., 2016*; *Vásquez-Restrepo et al., 2020*).

Of the seven currently recognized species of *Echinosaura*, four are known from Ecuador: *E. brachycephala*, *E. horrida*, *E. keyi* and *E. orcesi,* all from western humid tropical and subtropical areas (*Torres-Carvajal et al., 2021*).

In extreme northwestern Ecuador, specifically in Carchi province near the national border with Colombia, important remnants of native vegetation are still preserved due to inaccessibility and local conservation initiatives. During the last six years, we have focused on the study and conservation of this important biodiversity area, discovering several new species and many candidate species of small vertebrates that are still awaiting formal description (*Yánez-Muñoz et al., 2018*, *2020*; *Reyes-Puig et al., 2020*; *Brito et al., 2020*). Here, we describe a new species of *Echinosaura* recognized and delimited by a molecular phylogenetic analysis along with several types of morphological evidence, including cranial osteology. Accordingly, the first detailed description of the cranial morphology of *Echinosaura* based on CT scan data of the new species is presented.

## MATERIALS AND METHODS

### Ethics statement

We conducted this study under research permits MAE-DNB-CM-2016-0045 and MAE-DNB-CM-2019-0120, issued by the Ministerio del Ambiente del Ecuador. We followed the guidelines for use of live amphibians and reptiles in field research (*Beaupre et al., 2004*), compiled by the American Society of Ichthyologists and Herpetologists, the Herpetologists' League and the Society for the Study of Amphibians and Reptiles.

### Taxon sampling

We examined 36 specimens of *Echinosaura* (Appendix) housed in the following collections: División de Herpetología del Instituto Nacional de Biodiversidad (DHMECN), Quito, Ecuador; Museo de Zoología de la Pontificia Universidad Católica del Ecuador (QCAZ), Quito, Ecuador; Natural History Museum (BMNH), London, UK; University of Illinois Museum of Natural History (UIMNH), Illinois, USA; Muséum d'Histoire Naturelle de la Ville de Genève (MHNG), Geneva, Switzerland; Natural History Museum Vienna (NMW), Vienna, Austria; Zoological Research Museum Alexander Koenig (ZFMK), Bonn, Germany. We mapped locality records using ArcMap 10.5.1 (ESRI, Inc.) with a WGS84 datum and Universal Transverse Mercator conformal projection.

We adopted the unified species concept (*de Queiroz, 2007*) by considering different lines of evidence (*e.g.*, external morphology, osteology, hemipenial morphology, molecular phylogeny) as species delimitation criteria. We considered monophyly as additional evidence for recognizing populations as new species. The electronic version of this article in Portable Document Format (PDF) will represent a published work according to the International Commission on Zoological Nomenclature (ICZN), and hence the new names contained in the electronic version are effectively published under that Code from the electronic edition alone. This published work and the nomenclatural acts it contains have been registered in ZooBank, the online registration system for the ICZN. The ZooBank LSIDs (Life Science Identifiers) can be resolved, and the associated information viewed through any standard web browser by appending the LSID to the prefix http://zoobank.org/.

The LSID for this publication is: urn:lsid:zoobank.org:pub:9D513A0D-D676-4290-AE5D-A128B7DBE3F3.

**Field work**

We conducted field work in the foothill forests of Carchi province associated with several small river basins during a joint expedition of Instituto Nacional de Biodiversidad (INABIO) and Ecominga Foundation in three sectors of the Dracula Reserve: (1) Río Pailón Chico (0.982, −78.233; 1,224 m), 6–11 November 2017 and 19–25 August 2018; (2) Guapilal (0.891, −78.203; 1,640–1,750 m), 22–26 April 2019; (3) Cerro Oscuro (0.886, −78.190; 1,600 m), 4–6 July 2020; and (4) Bosque Comunal La Esperanza (0.954, −78.237; 1,439–1,623 m), 22–31 March 2021. We collected the specimens using visual encounter surveys and pitfall traps (*Fitzgerald, 2012*) and after photographing them alive, we euthanized the animals with benzocaine, extracted a sample of muscle tissue from each individual and preserved the samples in 95% ethanol. Subsequently, we fixed the specimens in 10% formalin, and preserved them in 75% ethanol.

**Morphological data and analysis**

Following *Vásquez-Restrepo et al. (2020)*, we recorded the following 12 morphometric characters with a digital caliper (Trupper, precision ±0.01 mm, rounded to 0.1 mm) using a 10x Boeco BS-80 stereo microscope: snout–vent length (SVL), measured from tip of snout to cloacal opening; trunk length (TRL), measured longitudinally from axilla to groin; tail length (TAL), measured from cloacal opening to tip of tail; snout length (SL), measured from tip of snout to anterior edge of eye; head length (HL), measured from tip of snout to anterior edge of tympanum; head width (HW), measured at level of greatest width; humerus length (HUM), measured from outer edge of elbow to edge of shoulder; forearm length (FAL), measured from edge of joint to base of hand; hand length (HND), measured from base of hand to tip of fourth finger; femur length (FEM), measured from groin to outer edge of knee; tibia length (TIB), measured from outer edge of knee to base of foot; foot length (FTL), measured from base of foot to tip of fourth toe. Except for males with everted hemipenes, we determined the sex by subcaudal incision, or by secondary sexual characteristics like the number of femoral pores. We used the terminology proposed by *Fritts, Almendariz & Samec (2002)*, *Köhler, Böhme & Schmitz (2004)* and *Vásquez-Restrepo et al. (2020)* for scale characters and measurements. We obtained data of color in life from field notes and photographs.

We extracted the right hemipenis of the male holotype (DHMECN 15208) from the recently collected specimen and then filled the extracted organ with blue-stained petroleum jelly and immersed it in a solution of alizarin red for two hours. Thereafter we washed the hemipenis to remove excess alizarin red and finally transferred it to 70% ethanol. Terminology for hemipenial structures follows *Uzzell (1973)*, *Savage (1997)* and *Nunes et al. (2012)*.

We obtained information on skeletal morphology and stomach contents from two specimens of the new species (DHMECN 15210, DHMECN 15208) by use of an X-ray in 2D (Faxitron X-ray LX60) and a micro-CT scanner (Bruker SkyScan 1173) in 3D, both

devices available at ZFMK. We placed the specimens in a plastic container and mounted them on styrofoam to avoid movements during scanning. We CT-scanned the specimens in 180° degrees using rotation steps of 0.3° degrees with a tube voltage of 35 kV and a tube current of 150 uA, without the use of a filter, at an image resolution of 39.3 μm. Scan duration was 30 min with an exposure time of 280 ms. We reconstructed the CT-datasets using N-Recon software (Bruker MicroCT) and rendered both scans in three dimensions with the program CTVox version 2.6 (Bruker MicroCT). Additionally, we rendered and segmented the skull of a female paratype (DHMECN 15210) in three dimensions to separate and color individual bones through the aid of Amira visualization software (FEI, Thermo Fisher Scientific). We provide a detailed description of the skull, vertebrae and pectoral girdle based mostly on data from the female paratype (DHMECN 15210). We omitted cartilage structures because micro-CT does not render cartilage. Osteological terminology follows *Evans (2008)*, *Jerez & Tarazona (2009)*, *Roscito & Rodrigues (2010)*, *Rodrigues et al. (2013)* and *Hernández Morales et al. (2019)*.

## DNA sequence data and phylogenetic analyses

We digested and extracted total genomic DNA from liver or muscle tissue using a guanidinium isothiocyanate extraction protocol. Using primers and amplification protocols from the literature (*Torres-Carvajal et al., 2016*), we generated DNA sequences of mitochondrial genes 12S, 16S, and ND4, as well as nuclear gene c-mos from two individuals of the new species described herein (GenBank accession numbers [DHMECN 14058 and 15208, respectively] MW525208–09 [12S], MW525211–12 [16S], MW512698–699 [ND4], and MW512700–701 [c-mos]). We retrieved all available GenBank sequences of these genes for other *Echinosaura* species, as well one sample per species of all clades ranked as genera within Cercosaurinae including "unnamed" clades (*Torres-Carvajal et al., 2016*).

We aligned DNA sequences using MAFFT (*Katoh & Standley, 2013*) under default settings in Geneious Prime 2020.2.2 (https://www.geneious.com), and translated ND4 and c-mos sequences into amino acids for confirmation of alignment. Our final concatenated data matrix contained 148 taxa and 1,935 characters and was partitioned by gene and codon position (*i.e.,* eight partitions total). We chose the best partitioning scheme using PartitionFinder v2.1.1 under the Bayesian Information Criterion (BIC), and the "greedy" algorithm with branch lengths of alternative partitions "linked" to search for the best-fit scheme (*Guindon et al., 2010*; *Lanfear et al., 2012*; *Lanfear et al., 2017*). We combined the genes into a single dataset with six partitions: (i) 12S, 16S [GTR + I + G]; (ii) 1st codon position of ND4 [GTR + I + G]; (iii) 2nd codon position of ND4 [GTR + I + G]; (iv) 3rd codon position of ND4 [GTR + G]; (v) 3rd codon position of c-mos [K80 + G]; and (vi) 1st and 2nd codon positions of c-mos [HKY + G]. We ran a maximum likelihood analysis in RAxML v8.2.10 (*Stamatakis, 2014*) under the GTRGAMMA model, while we assessed nodal support with the rapid bootstrapping (BS) algorithm (*Stamatakis, Hoover & Rougemont, 2008*) on 1,000 replicates. We also ran a Bayesian analysis in MrBayes v3.2.7 (*Ronquist et al., 2012*), with all parameters unlinked between partitions (except topology and branch lengths) and rate variation (prset ratepr = variable) invoked. We set four independent runs, each with four MCMC chains, for $10^7$ generations, sampling every

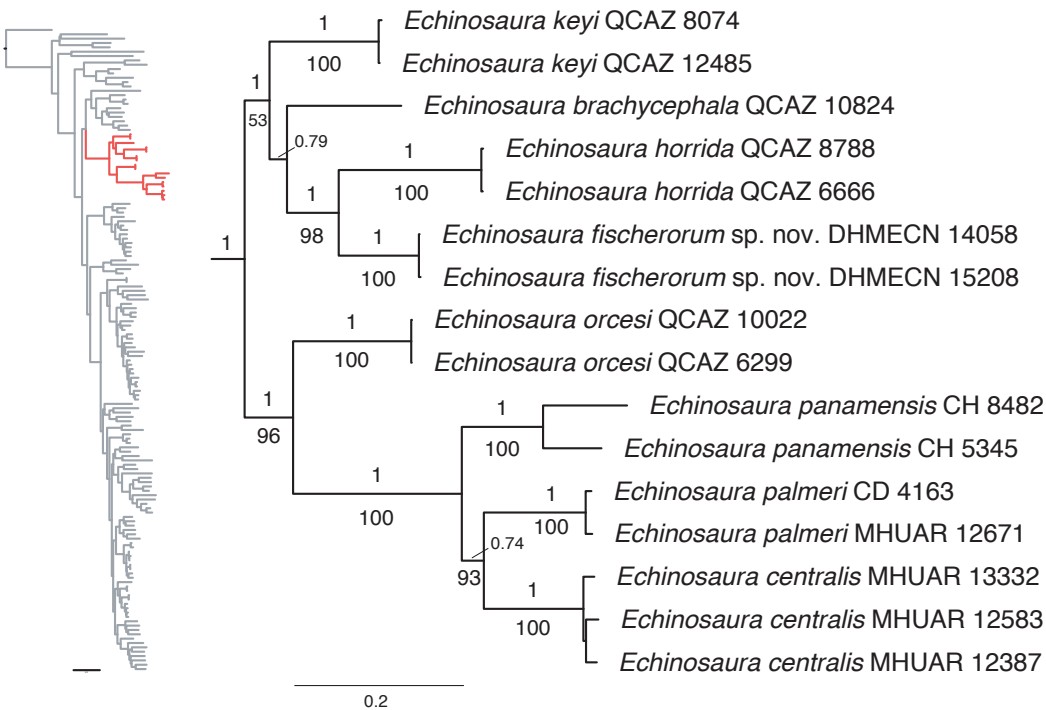

**Figure 1  Phylogeny of Cercosaurinae, with a close-up of *Echinosaura*.** Maximum clade credibility tree obtained from a Bayesian analysis of 148 taxa and 1,935 characters. Numbers above branches are Bayesian posterior probability (PP) values and those below branches are ML bootstrap support (BS) values.

10,000 generations. We calculated posterior probabilities (PP) on a Maximum Clade Credibility Tree in TreeAnnotator (*Rambaut & Drummond, 2019*) and rooted the trees with *Alopoglossus* (*Castoe, Doan & Parkinson, 2004*; *Goicoechea et al., 2016*; *Pellegrino et al., 2001*). We executed phylogenetic analyses in the CIPRES Science Gateway (*Miller, Pfeiffer & Schwartz, 2010*). In addition, we calculated pairwise genetic distances for ND4 and 16S genes among species of *Echinosaura* using DIVEIN (*Deng et al., 2010*).

## RESULTS

### Phylogenetic relationships

Maximum likelihood and Bayesian analyses resulted in nearly identical topologies (Fig. 1). Both analyses support monophyly of the new species of *Echinosaura* described in this paper. The monophyly of *Echinosaura* is maximally supported (BS = 100, PP = 1). There is a basal split into two clades, one of which (BS = 96, PP = 1) contains *E. orcesi* as sister to (*E. panamensis*, (*E. centralis*, *E. palmeri*)). The second clade (BS = 53, PP = 1) includes, following branching order, *E. keyi*, *E. brachycephala* (sister to *E. keyi* in the ML tree), and the new species described below as sister to *E. horrida* with strong support (BS = 98, PP = 1). Uncorrected pairwise genetic distances for ND4 between *Echinosaura* sp. nov. and other congenerics range between 0.20, with both *E. horrida* and *E. keyi*, and 0.35 with *E. centralis*. Distances for 16S range from 0.06 with *E. brachycephala* to 0.14 with *E. panamensis*.

## Systematic accounts

**Squamata** Oppel, 1811
**Gymnophthalmidae** Merrem, 1820
*Echinosaura* Boulenger, 1890

### *Echinosaura fischerorum* sp. nov.

LSID: urn:lsid:zoobank.org:act:F8DCFE99-4862-4476-9A5C-EB684D0CD73A

**Proposed standard Spanish name:** Lagartijas espinosas de los Fischer
**Proposed standard English name:** Fischers' Spiny Lizards
**Holotype (Figs. 2, 3, 4, 5, 6, 7).** DHMECN 15208, adult male, from Reserva Dracula, Sector El Guapilal (0.891, −78.203; 1,689 m), Carchi Province, Ecuador, collected on 22 April 2019 by Juan P. Reyes-Puig, Daniela Franco and Héctor Yela.
**Paratypes (Figs. 4, 5, 6, 8, 9, 10).** DHMECN 15209 and DHMECN 15211, adult males, DHMECN 15210, adult female, all with same data as holotype; DHMECN 14058 and DHMECN 14060, adult females, DHMECN 14059 and DHMECN 14061, juvenile females, all from Reserva Dracula, Sector El Pailón Chico (0.983, −78.296; 1,495 m), collected on 8 November 2017 by Mario H. Yanez-Muñoz, Juan P. Reyes-Puig and Fausto Recalde; DHMECN 12767, juvenile female from Cerro Oscuro, Reserva Dracula (0.886, −78.190; 1,600 m) collected on June 2015 by Héctor Yela; DHMECN 16109, adult female and DHMECN 16110, adult male, from Comunidad La Esperanza (0.954, −78.237; 1,623 m), collected on 28 March 2021 by Mario H. Yánez-Muñoz, Juan P. Reyes-Puig and Miguel A. Urgiles-Merchán.
**Diagnosis**. The new species can be distinguished from all congeners by the combination of the following characteristics: (1) snout pointed; (2) internasal single; (3) frontonasals paired; (4) frontal single; (5) frontoparietal paired; (6) supraoculars three, large; (7) supralabials five; (8) infralabials four; (9) postmental single; (10) chin shields enlarged, in one pair; (11) dorsum with a vertebral row of paired, enlarged, keeled scales; (12) two paravertebral series of short oblique rows of projecting scales, with scales increasing in size posteriorly on each row so that the most posterior scale of each row is a greatly enlarged, projecting spine; (13) spiny scales forming oblique lines on body flanks; (14) ventral scales squared, keeled; (15) subdigital lamellae on fourth finger 14–18; (16) subdigital lamellae on fourth toe 24–27; (17) femoral pores per hind limb in males 7–9; (18) dorsal surface of tail with two longitudinal rows of enlarged keeled scales that are more conspicuous on the anterior half of tail; (19) subcaudals per caudal segment three (anterior third of tail excluded); (20) tip of snout with creamy orange marks; (21) dorsal background dark brown with creamy orange paravertebral blotches extending onto anterior end of tail; (22) intense orange nuchal spines; (23) base of tail dorsally with a pair of pale orange blotches; (24)

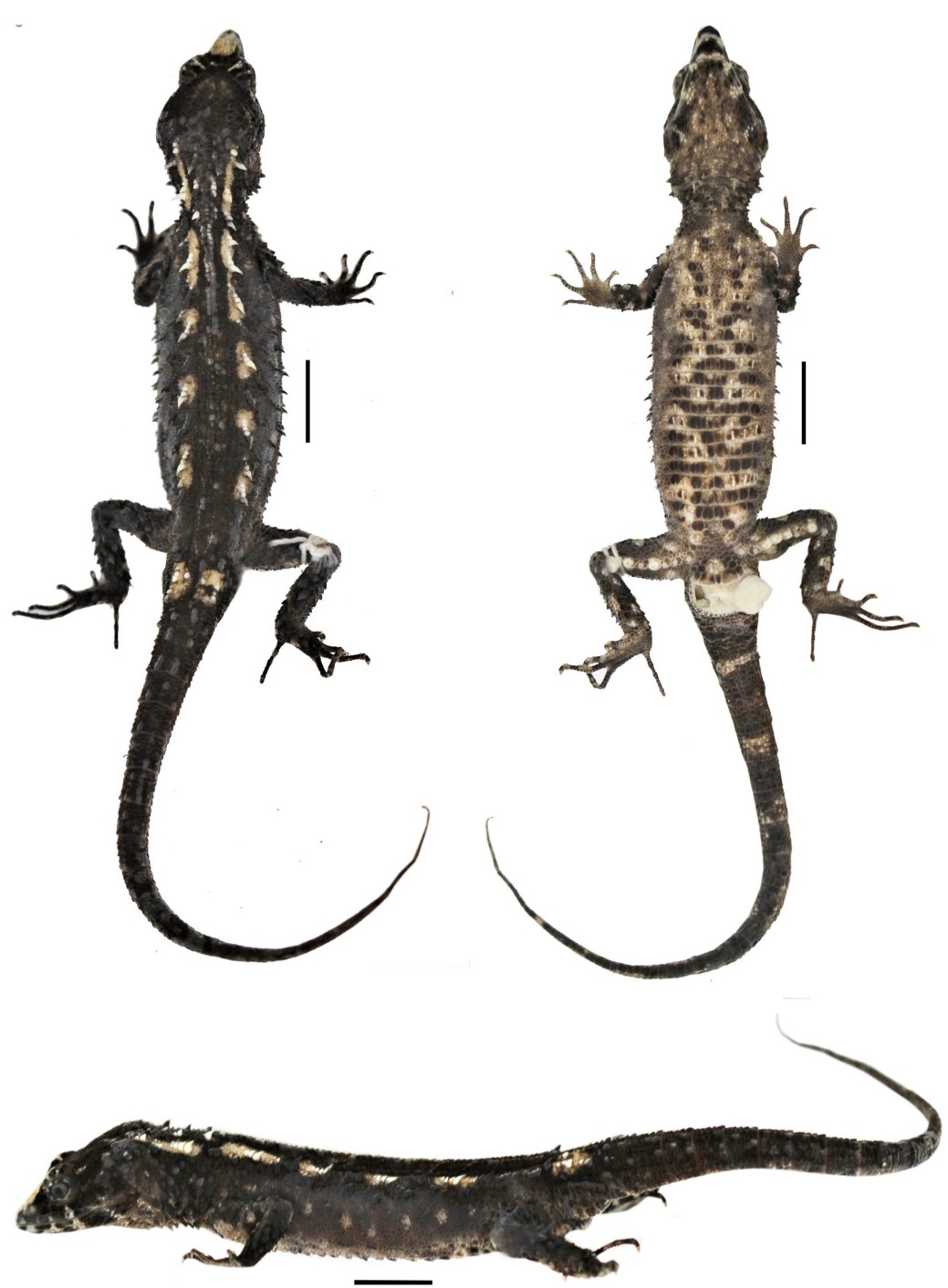

**Figure 2** Holotype of *Echinosaura fischerorum* sp. nov. (male, DHMECN 15208) in dorsal (upper left), ventral (upper right), and lateral (bottom) views. Scale bars = 10 mm. Photographs by MYM.

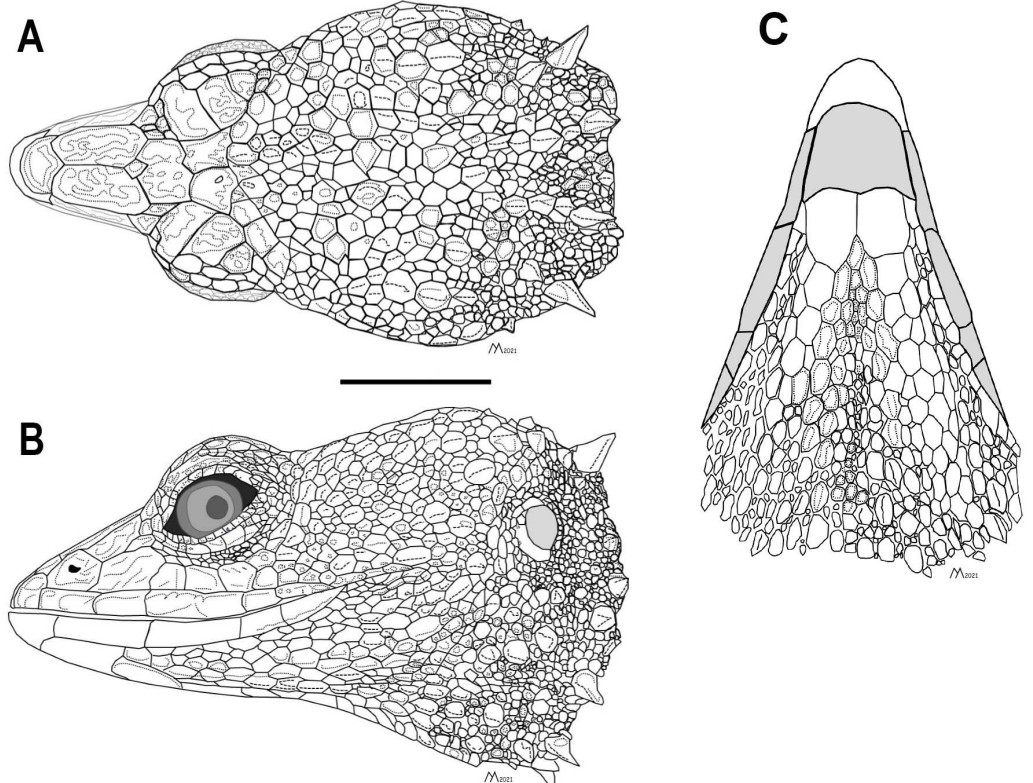

**Figure 3** **Head of the holotype of *Echinosaura fischerorum* sp. nov. (male, DHMECN 15208).** (A) Dorsal view, (B) lateral view and (C) ventral view. Scale bar = 10 mm. Infralabial and mental scales are shown in gray in ventral view. Illustrations by MYM.

venter gray, marmorated with black and dark brown; (25) femoral pores yellowish cream; (26) premaxillary tooth loci 11–12; (27) maxillary tooth loci 17–18; (28) dentary tooth loci 22–23; (29) nasal bones in medial contact along most of their length; (30) postfrontal expands posteriorly to form part of the anterior border of the supratemporal fenestra.

**Comparison with similar species (Figs. 11, 12, 13, 14).** *Echinosaura fischerorum* sp. nov. differs from all *Echinosaura* species in having keeled enlarged dorsal scales forming a paired vertebral row, two paravertebral series of short oblique rows of projecting scales, and a pair of spine-like scales on each side in temporal and nuchal regions (see Table 1).

*Echinosaura fischerorum* sp. nov. most closely resembles *E. horrida* (character states in parentheses), but can be distinguished from this species in having a wide postmental in contact with anterior 3/4 of first infralabial (Fig. 3C; postmental longer than wide, in contact with first and second infralabials); keeled (smooth) enlarged dorsal scales forming a paired vertebral row; scales on each side of vertebral row heterogeneous in size (granular, homogeneous in size), with enlarged scales forming chevrons (Figs. 12A, 12B); sharp (blunt) spiny scales throughout body, forearms and legs; striated (smooth) dorsal and lateral head scales; keeled (smooth) parietals; nasal bones in medial contact along most of their length (distinctly separated, resulting in contact between premaxilla and frontal

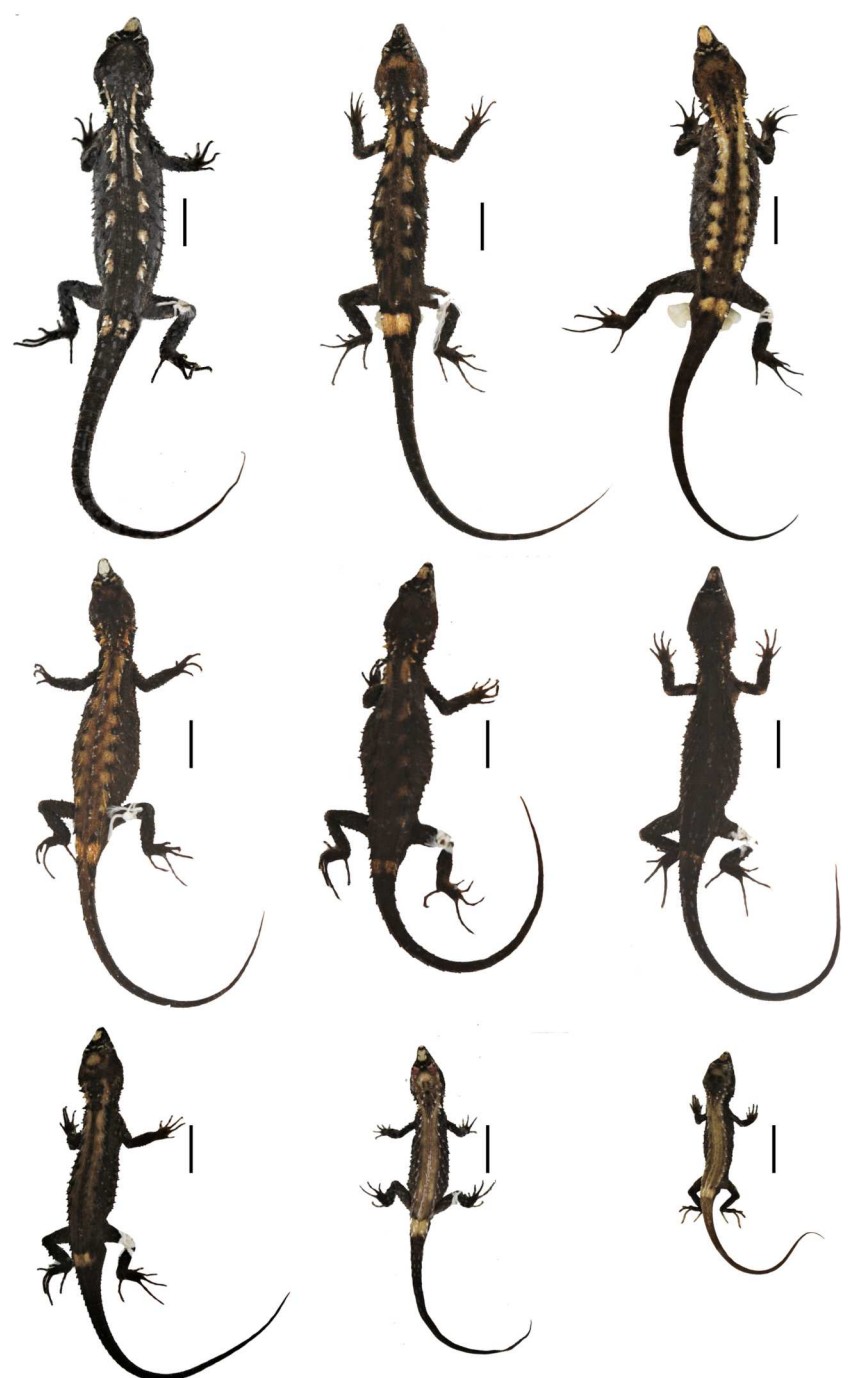

**Figure 4  Preserved type specimens of *Echinosaura fischerorum* sp. nov. showing color variation in dorsal view.** Top (males): DHMECN 15208 (Holotype), DHMECN 15209 (Paratype), DHMECN 15211 (Paratype); middle (females): DHMECN 15210 (Paratype), DHMECN 14058 (Paratype), DHMECN 14060 (Paratype); bottom (juveniles): DHMECN 14061 (Paratype), DHMECN 12767 (Paratype), DHMECN 14059 (Paratype). Scale bars = 10 mm. Photographs by MYM.

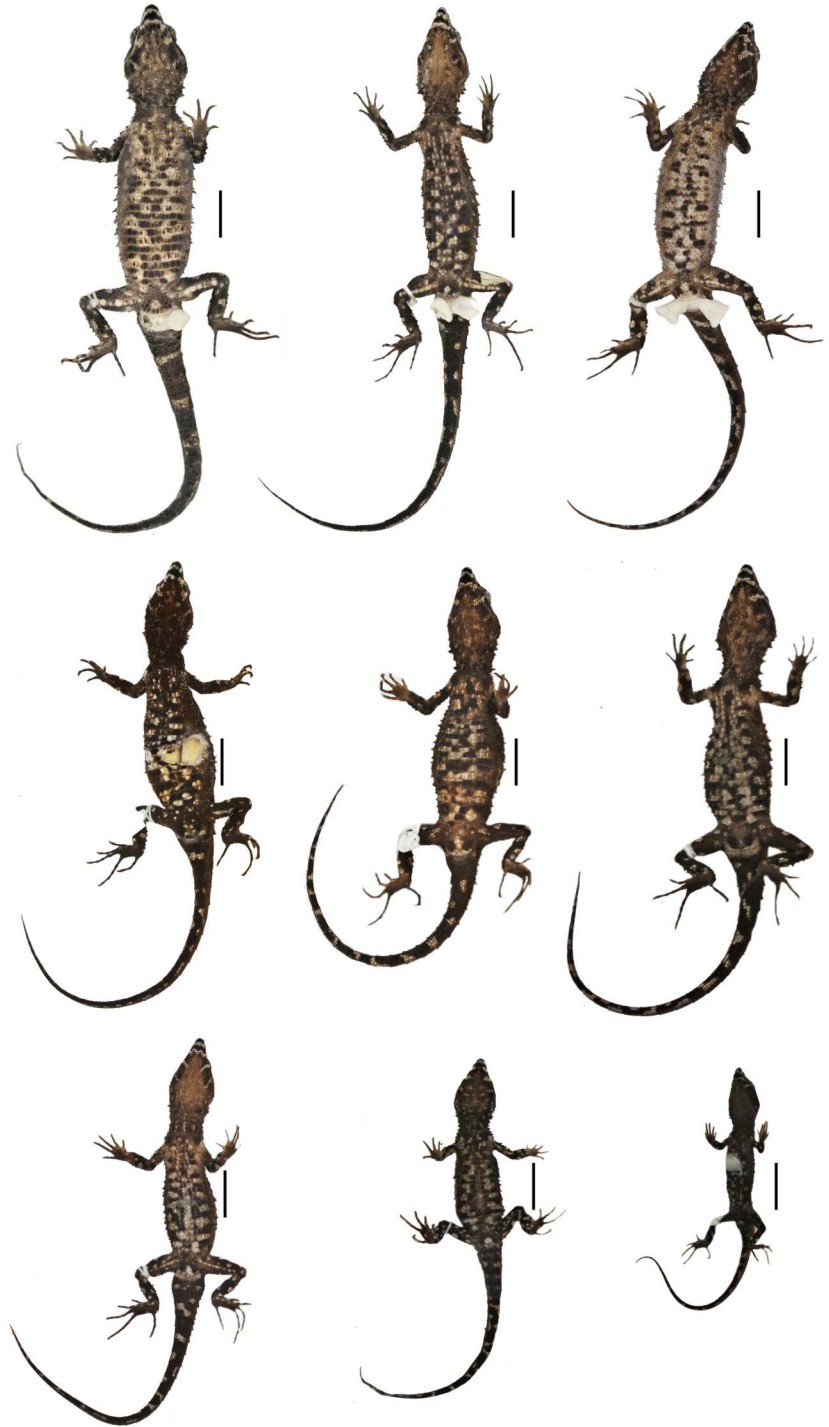

**Figure 5 Preserved type specimens of *Echinosaura fischerorum* sp. nov. showing color variation in ventral view.** Top (males): DHMECN 15208 (Holotype), DHMECN 15209 (Paratype), DHMECN 15211 (Paratype); middle (females): DHMECN 15210 (Paratype), DHMECN 14058 (Paratype), DHMECN 14060 (Paratype); bottom (juveniles): DHMECN 14061 (Paratype), DHMECN 12767 (Paratype), DHMECN 14059 (Paratype). Photographs by MYM.

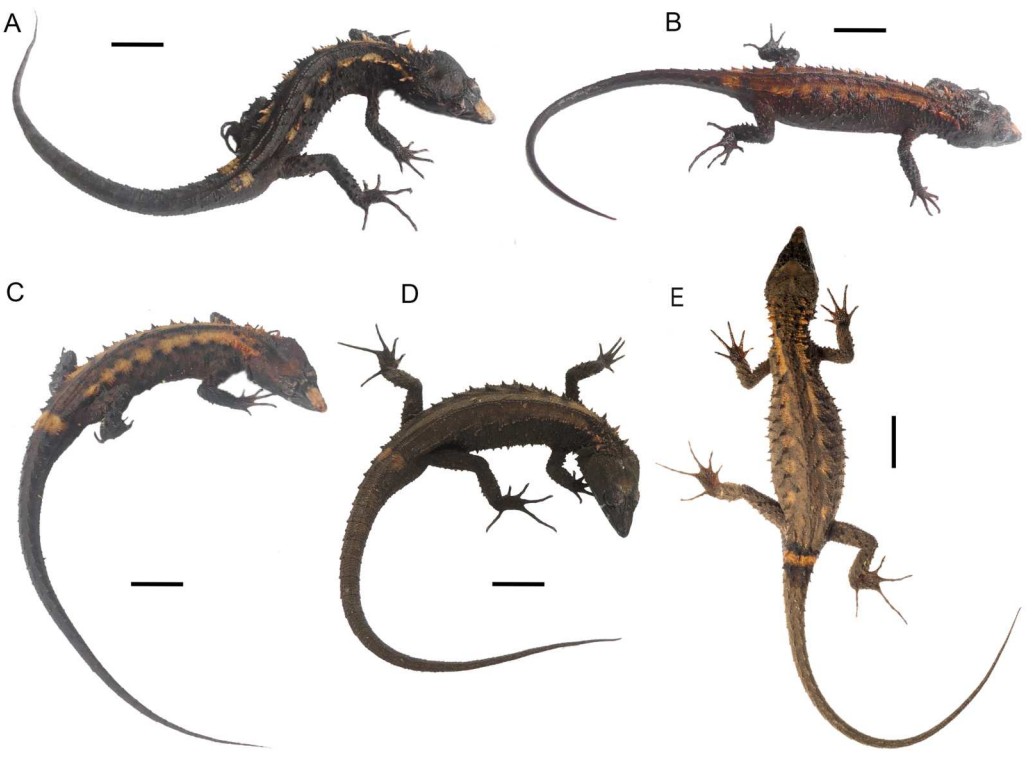

**Figure 6 Live type specimens of *Echinosaura fischerorum* sp. nov. showing color variation in dorsal view.** (A) DHMECN 15208, male (Holotype); (B) DHMECN 15209, male; (C) DHMECN 15211, male; (D) DHMECN 16109, female; (E) DHMECN 16110, male. Scale bars = 10 mm. Photographs by JPRP (A, B, C); MAUM (D); Andrew Better (E).

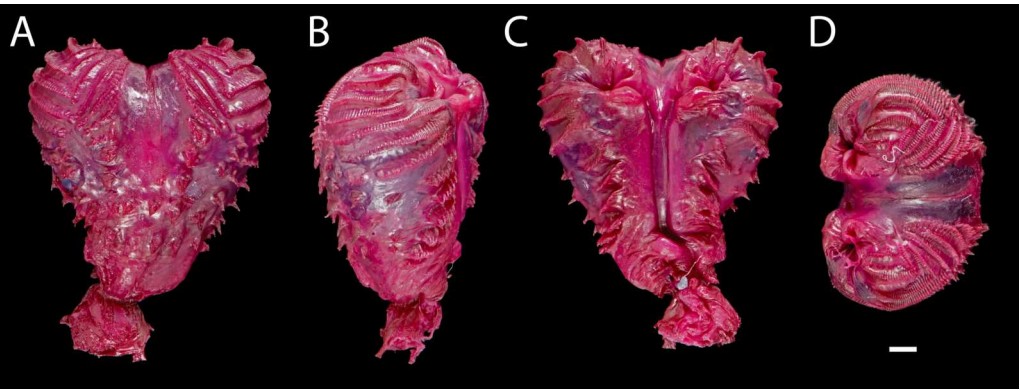

**Figure 7 Right hemipenis of male holotype of *Echinosaura fischerorum* sp. nov. (DHMECN 15208, Holotype) in (A) asulcate, (B) lateral, (C) sulcate, and (D) apical views.** Scale bar = 1 mm. Photographs by Morris Flecks.

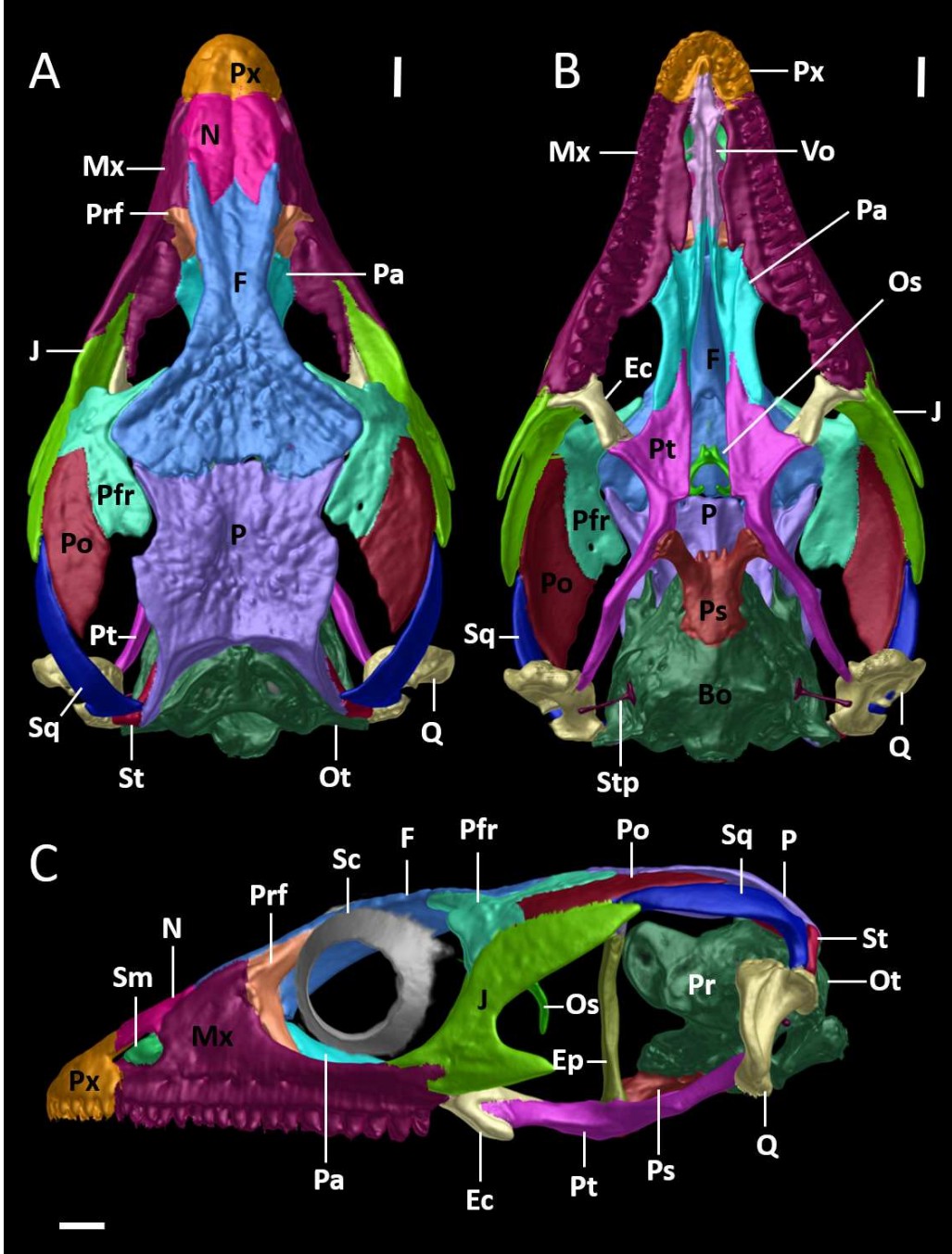

**Figure 8  Cranial skeleton of female paratype (DHMECN 15210) of *Echinosaura fischerorum* sp. nov.**
The skull is shown without mandible in (A) dorsal, (B) ventral, (C) lateral views. Bo, basioccipital; Ec, ectopterygoid; Ep, epipterygoid; F, frontal; J, jugal; Mx, maxilla; N, nasal; Os, orbitosphenoid; Ot, otoccipital; P, parietal; Pa, palatine; Pfr, postfrontal; Po, postorbital; Pr, prootic; Prf, prefrontal; Ps, parabasisphenoid; Pt, pterygoid; Px, premaxilla; Q, quadrate; Sc, scleral ossicles; Sm, septomaxilla; So, supraoccipital; Sq, squamosal; St, supratemporal; Stp, stapes; Vo, vomers. Scale bars = 1 mm.

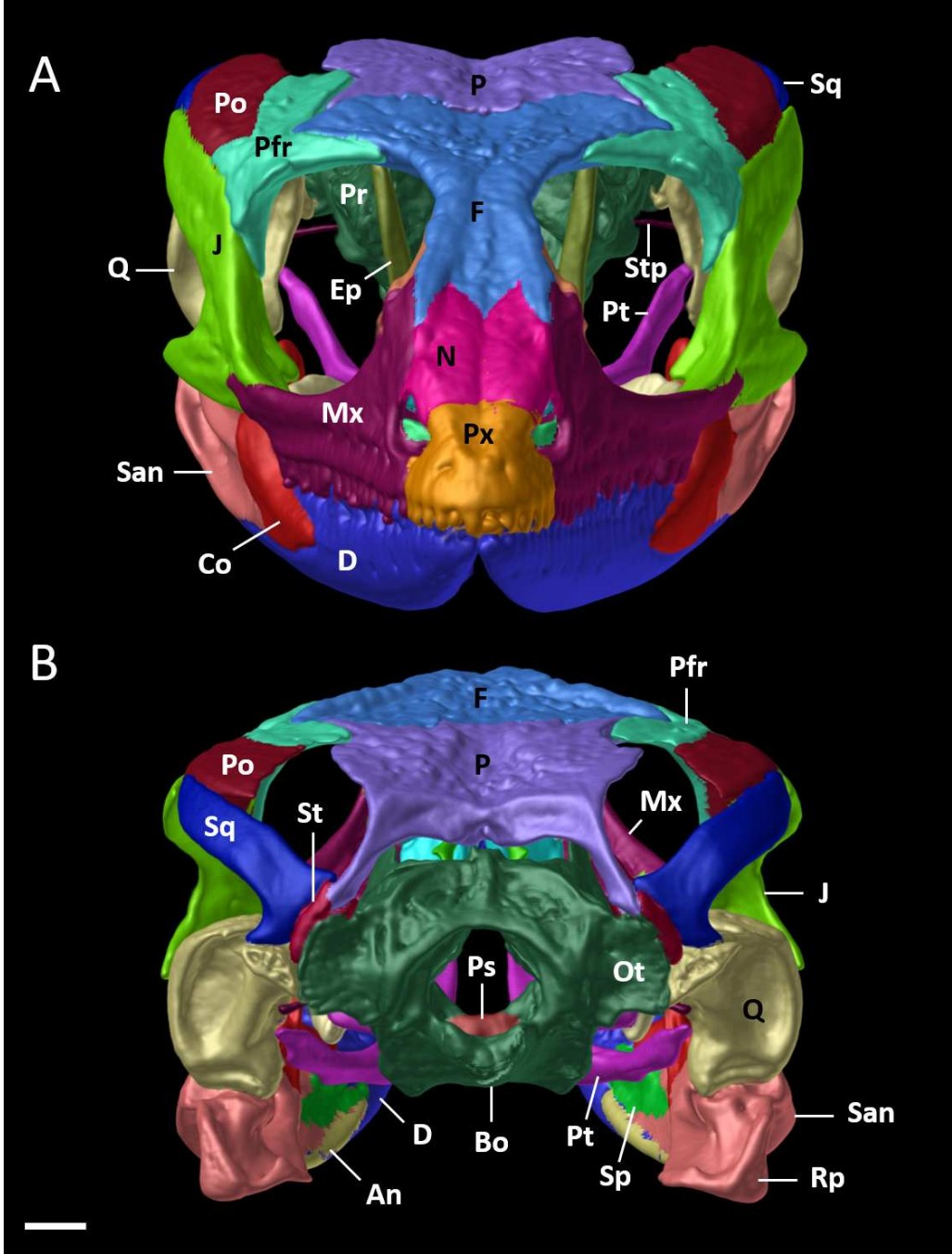

**Figure 9 Cranial skeleton of female paratype (DHMECN 15210) of *Echinosaura fischerorum* sp. nov.**
The skull is shown in (A) anterior and (B) posterior views. An, angular; Bo, basioccipital; Co, coronoid; D, dentary; Ep, epipterygoid; F, frontal; J, jugal; Mx, maxilla; N, nasal; Ot, otoccipital; P, parietal; Pfr, post-frontal; Po, postorbital; Pr, prootic; Ps, parabasisphenoid; Pt, pterygoid; Px, premaxilla; Rp, retroarticular process; Q, quadrate; San, surangular; Sp, splenial; Sq, squamosal; St, supratemporal; Stp, stapes. Scale bar = 1 mm.

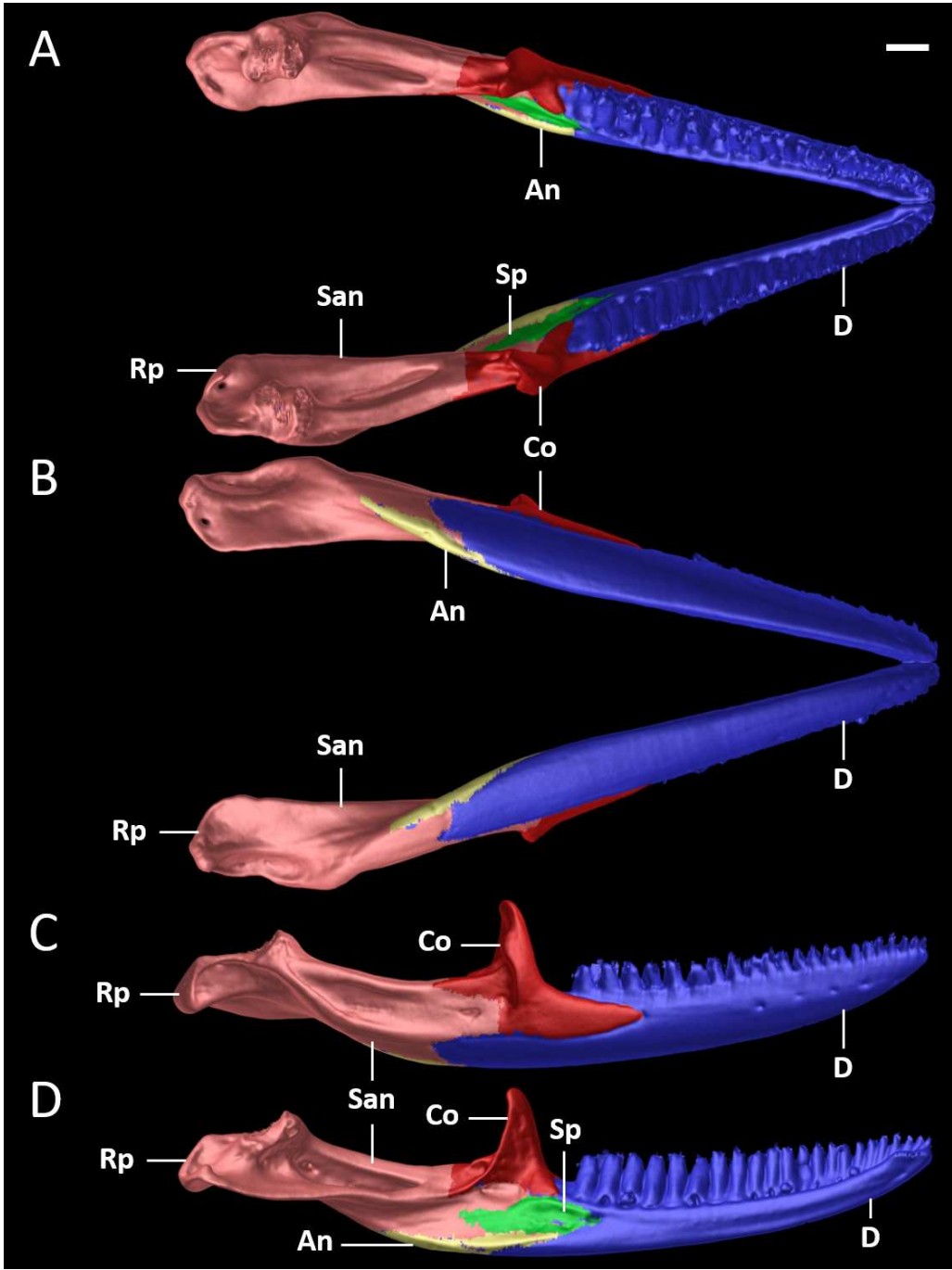

**Figure 10  Mandible of female paratype (DHMECN 15210) of *Echinosaura fischerorum* sp. nov. in (A) dorsal, (B) ventral, (C) lateral, and (D) medial views.** An, angular; Co, coronoid; D, dentary; Rp, retroarticular process; San, surangular; Sp, splenial. Scale bar = 1 mm.

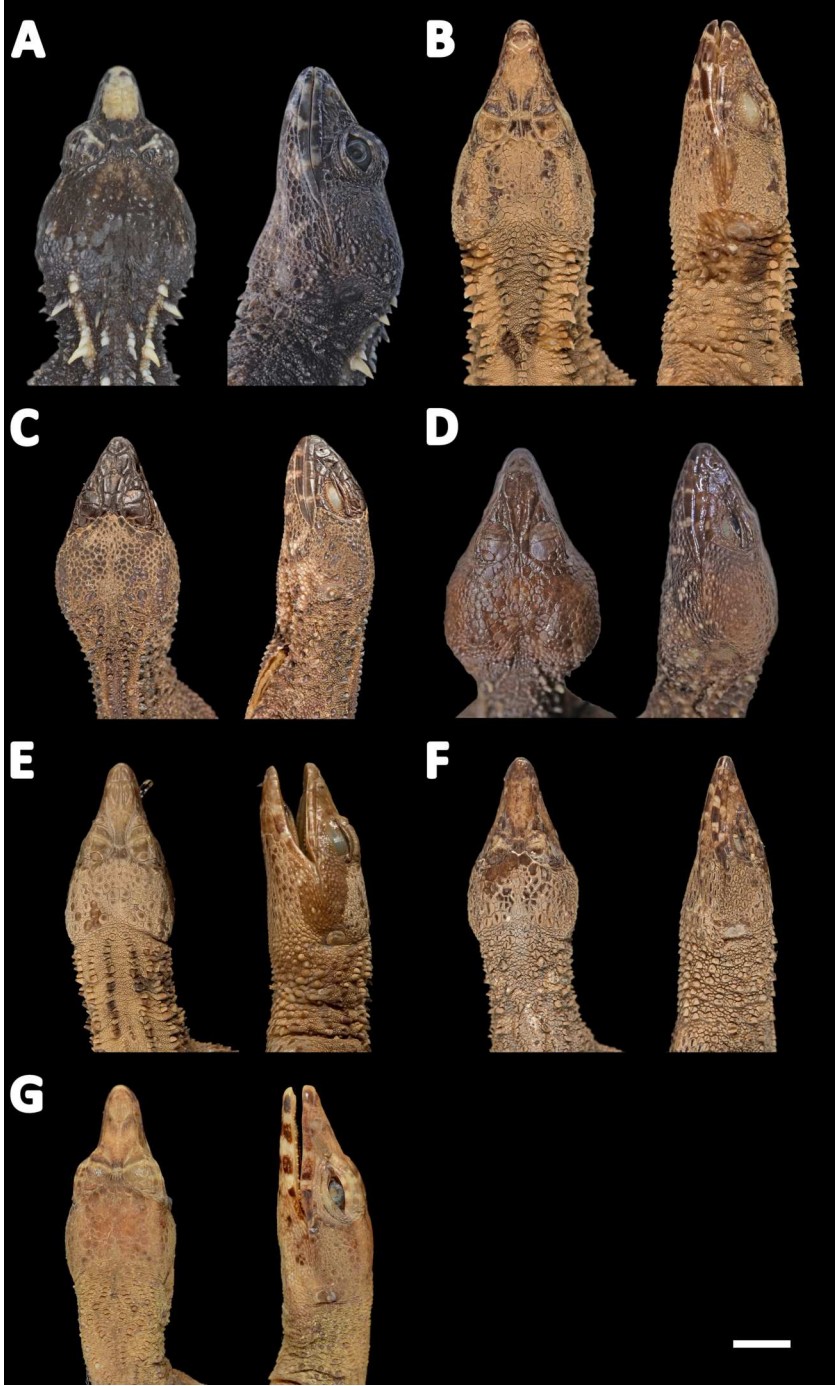

**Figure 11** **Heads of seven species of *Echinosaura* in dorsal (left) and lateral (right) views.** (A) *Echinosaura fischerorum* sp. nov. (DHMECN 15208, Holotype), (B) *E. horrida* (ZFMK 43763), (C) *E. brachycephala* (ZFMK 46370, Paratype), (D) *E. keyi* (UIMNH 80451, Holotype), (E) *E. palmeri* (BMNH 1923.10.12.14), (F) *E. panamensis* (ZFMK 52200), (G) *E. orcesi* (NMW 32000:1, Paratype). Scale bar = 5 mm. Photographs by MYM (A); Morris Flecks (B, C, E–G); Chris Phillips (D).

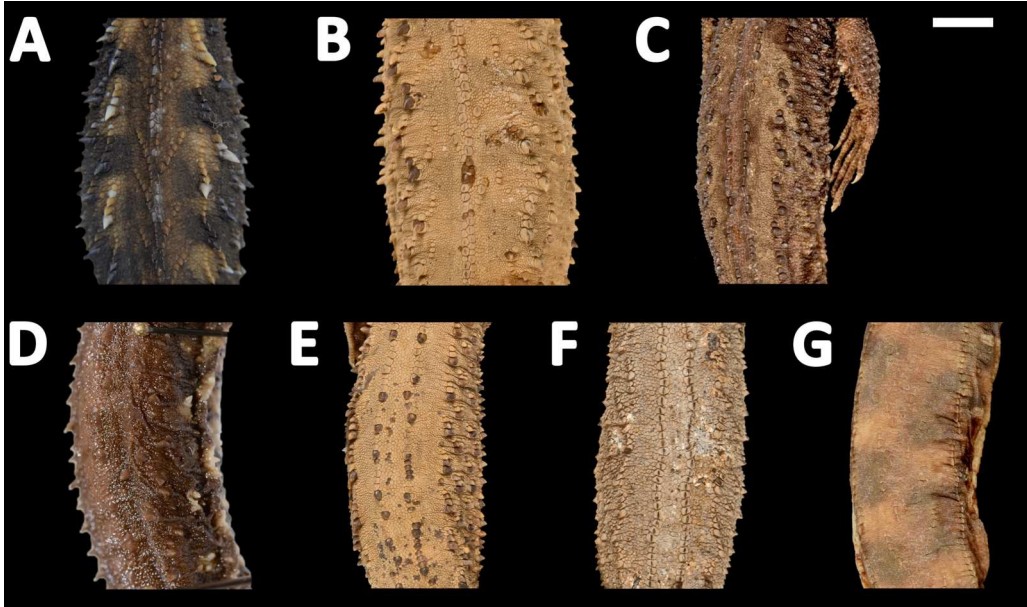

**Figure 12** **Midbody of seven species of *Echinosaura* in dorsal view.** (A) *Echinosaura fischerorum* sp. nov. (DHMECN 15209, Paratype), (B) *E. horrida* (ZFMK 43763), (C) *E. brachycephala* (ZFMK 46370, Paratype), (D) *E. keyi* (UIMNH 80451, Holotype), (E) *E. palmeri* (BMNH 1923.10.12.14), (F) *E. panamensis* (ZFMK 52200), (G) *E. orcesi* (NMW 32000:2, Paratype). Scale bar = 5 mm. Photographs by MYM (A); Morris Flecks (B, C, E–G); Chris Phillips (D).

bone); postfrontal bone expands posteriorly to form part of the anterior border of the supratemporal fenestra (postfrontal excluded from the supratemporal fenestra by contact of postorbital with parietal). Although our sampling size is small, *E. fischerorum* sp. nov. seems to be smaller (maximum SVL = 68 mm) than *E. horrida* (maximum SVL = 86 mm).

**Description of adult male holotype (Figs. 2, 3).** Rostral single, in contact with internasal posteriorly, nasals and first supralabials posterolaterally; internasal single, wider than long, rugose, in contact with rostral anteriorly, nasals laterally, and frontonasals posteriorly; nasals pentagonal, in contact with rostral anteriorly, internasal and frontonasal dorsally, frenocular and lorilabial posteriorly, and first supralabial ventrally; frontonasals paired, rugose, nearly rectangular and twice as long as wide, with posterior edges forming an obtuse angle, in contact with internasal anteriorly, pre-supraocular scales and frontal posteriorly, and nasal and frenocular ventrally; frontal single, rugose, subpentagonal, wider than long, in contact with frontonasals anteriorly, pre-supraocular scales and first supraocular scales laterally, and frontoparietals posteriorly; supraoculars three; first supraocular larger than others, in contact with frontal and pre-supraocular scales anteriorly, frontoparietal medially, supercilliaries laterally, and second supraoccipital posteriorly; lower eyelid with unpigmented palpebral disc divided into three large scales; frenocular large, rectangular, about twice as long as tall, in contact with nasal anteriorly, frontonasal dorsally, lorilabial ventrally, preoculars and first pre-supraocular posteriorly; lorilabial in contact with nasal anteriorly, first and second supralabials ventrally, frenocular and preoculars dorsally, and

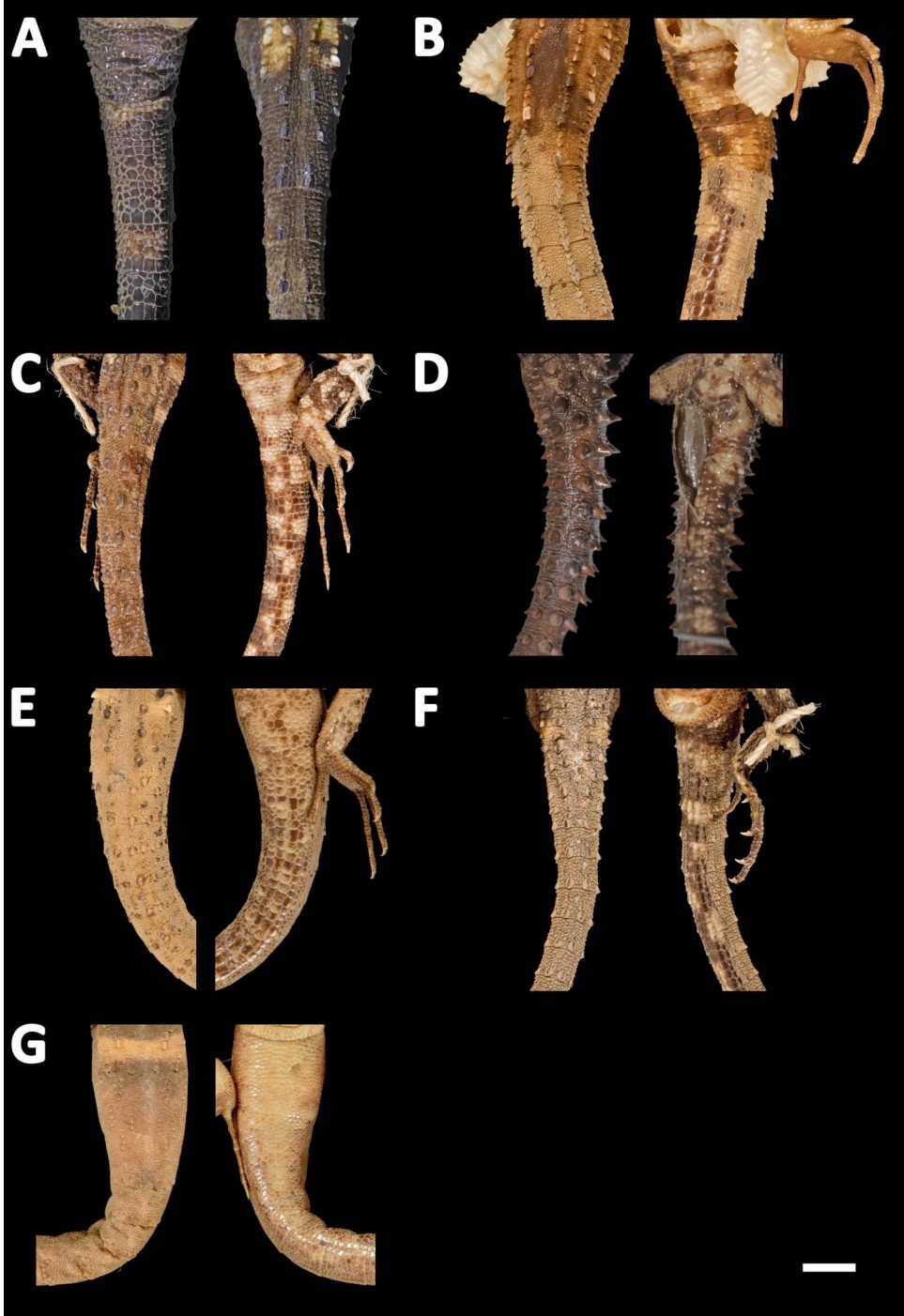

**Figure 13 Tails of seven species of *Echinosaura* in dorsal (left) and ventral (right) views.** (A) *Echinosaura fischerorum* sp. nov. (DHMECN 15208, Holotype), (B) *E. horrida* (ZFMK 43763), (C) *E. brachycephala* (ZFMK 46370, Paratype), (D) *E. keyi* (UIMNH 80451, Holotype), (E) *E. palmeri* (BMNH 1923.10.12.14), (F) *E. panamensis* (ZFMK 52200), (G) *E. orcesi* (NMW 32000:2, Paratype). Scale bar = 5 mm. Photographs by MYM (A); Morris Flecks (B, C, E–G); Chris Phillips (D).

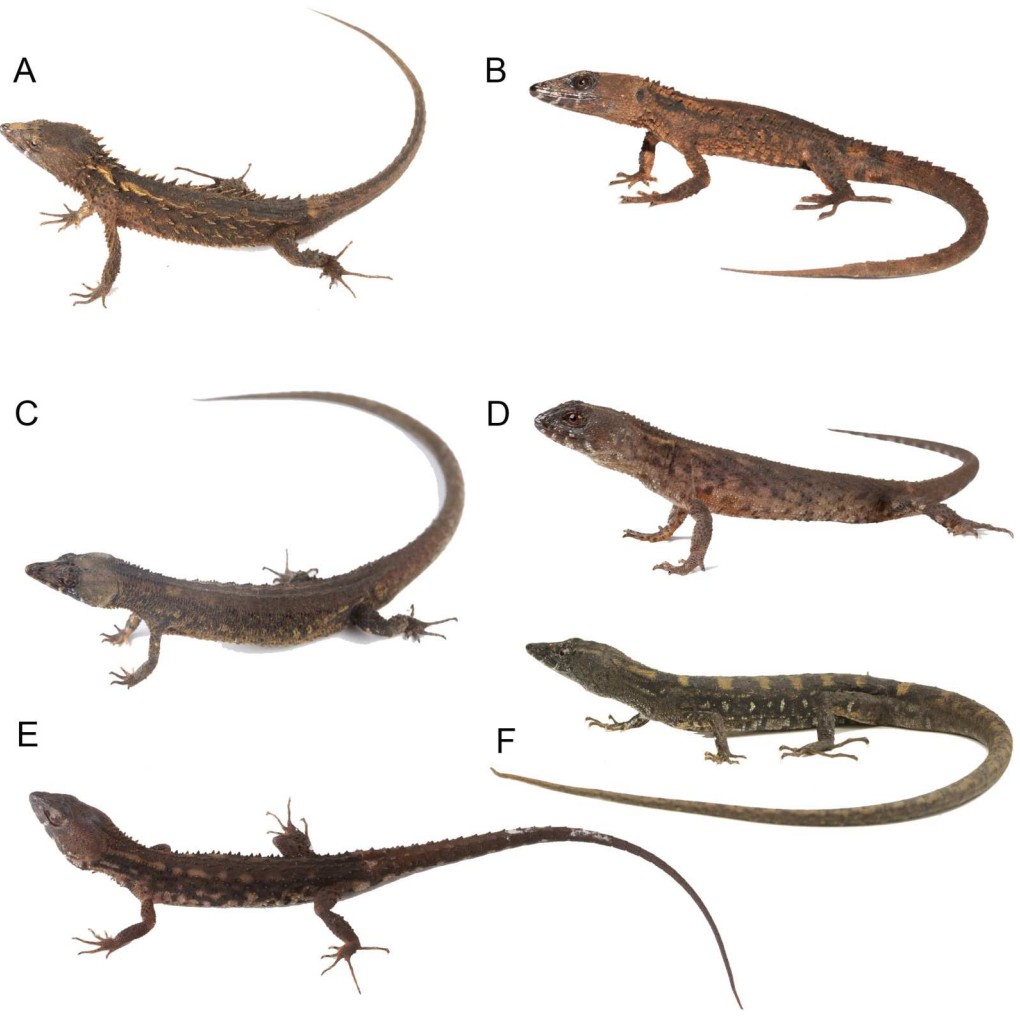

**Figure 14 Live specimens of six species of *Echinosaura*.** (A) *E. fischerorum* sp. nov., male (no voucher), Reserva Manduriacu, Pichincha, Ecuador; (B) *E. horrida*, male (DHMECN 13913), Cerro Zapallo, Esmeraldas, Ecuador; (C) *E. palmeri*, male (no voucher), Timbiqui, Colombia; (D) *E. brachycephala*, male (QCAZ 11911), Reserva Ecológica Bosque Nublado Santa Lucía, Pichincha, Ecuador; (E) *E. keyi*, male (QCAZ 8074), Reserva Otokiki, Esmeraldas, Ecuador; (F) *E. orcesi*, male (QCAZ 15026), Reserva Tesoro Escondido, Esmeraldas, Ecuador. Photographs by MYM (B); Jaime Culebras (A, C); Diego Quirola (D, E); Santiago R. Ron (F).

first subocular posteriorly; suboculars in 2–5 longitudinal rows, of which the ventralmost contains five enlarged scales in contact with supralabials III–V; frontoparietals two, similar in size to frontal when combined, contacting frontal anteriorly, first and second supraocular laterally, and scales of parietal region posteriorly; parietal region covered with small irregular scales; one pair of spine-like temporals on each side; supralabials five; infralabials four; mental wider than long, in contact with postmental posteriorly and first infralabials laterally; postmental trapezoidal, in contact with first infralabials laterally and one pair of chin shields and two small lateral scales posteriorly; chin shields in one pair,

about twice as long as wide, separated from infralabials by small scales; gular region with five transverse rows of 5–7 large spiny scales that extend to lower part of tympanum.

Dorsum with a vertebral row of paired, enlarged, keeled scales, and two paravertebral series of short oblique rows of projecting scales, with scales increasing in size posteriorly on each row so that the most posterior scale forms a greatly enlarged, projecting spine; scales between vertebral row and paravertebral series of short rows heterogeneous in size, with enlarged scales forming oblique rows that converge medially to form chevrons; body flanks with seven oblique rows of spine-like scales, which are separated from each other by small, irregular, tubercular scales; pectoral region with four longitudinal rows of enlarged, keeled ventral scales, with rounded edges, intermixed with many smaller scales; eight longitudinal rows of squared, enlarged, keeled ventral scales on belly, not intermixed with smaller scales.

Limbs pentadactyl; digits clawed; subdigital lamellae on fourth finger single (second subdigital on left side divided), 16 on left side and 14 on right side; subdigital lamellae on fourth toe 24 on both sides; dorsal surface of tail with two longitudinal rows of enlarged keeled scales in five to six pairs per caudal segment, increasing in size posteriorly on each segment; caudal longitudinal rows separated from each other by 3–7 small scales; four pairs of subcaudals per autotomic segment on anterior third of tail; three pairs of subcaudals per autotomic segment on posterior two thirds.

**Coloration of holotype in life (Fig. 6A).** The holotype has a dark brown dorsal background with creamy orange paravertebral blotches extending onto anterior end of tail; flanks with grayish cream circular marks; parietal region brown orange; temporal region and posterior border of mandible with scattered white scales; tip of snout with creamy orange marks; first supraocular with diagonal grayish cream line; supercilliaries with 2–3 pale marks; supralabials with a pair of grayish cream bars; nuchal spines intense orange; some spines along paravertebral rows creamy orange; dorsally, the base of the tail with a pair of pale orange blotches; venter gray, marmorated with black and dark brown; femoral pores yellowish cream; round, cream blotch on supra-anal region; limbs dark brown with grayish cream marks; iris copper red.

**Coloration of holotype after 1–5 years in preservative (Figs. 2, 4, 5).** The holotype is dorsally brown, with six pairs of paravertebral cream blotches; base of tail with broad transverse, yellowish cream bar; tip of snout with cream nasal and internasal scales; lips and chin with cream bands separated by black interspaces; ventral surface of body and limbs dark brown with cream marbling; femoral pores pale gray.

**Measurements of the holotype (in mm).** SVL = 69 mm; TAL = 89.5 mm; TRL = 33.6 mm; SL = 13.7 mm; HL = 20.1 mm; HW = 13.2 mm; HUM = 9.7 mm; FAL = 9.1 mm; HND = 9.3 mm; FEM = 12.5 mm; TIB = 13.7 mm; FTL = 15.7 mm

**Variation (Figs. 4, 5, 6).** Tables 2 and 3 show the intraspecific variation in morphometrics and lepidosis, most of which was noted after 1-5 years in preservation (Figs. 4, 5). In some male specimens, the paravertebral cream spots are fused together, sometimes forming continuous paravertebral stripes (DHMECN 15211); in females, both the paravertebral blotches and the dorsal transverse bar on base of tail are faint (DHMECN 14058) or absent (DHMECN 14060), whereas juveniles have a pair of light paravertebral stripes

Yánez-Muñoz et al. (2021), *PeerJ*, DOI 10.7717/peerj.12523

**Table 1** **Variation in selected morphological characters of the different species of *Echinosaura*.**

| SPECIES | *E. fischerorum* sp nov. (*n* = 9) | *E. brachycephala* (*n* = 25) | *E. centralis* (*n* = 39) | *E. horrida* (*n* = 27) | *E. keyi* (*n* = 14) | *E. orcesi* (*n* = 4) | *E. palmeri* (*n* = 44) | *E. panamensis* (*n* = 10) |
|---|---|---|---|---|---|---|---|---|
| Max. SVL males (mm) | 69 (*n* = 3) | 72 (*n* = 10) | 75 | 86 (*n* = 14) | 80 (*n* = 7) | 70 (*n* = 2) | 80 | 71 (*n* = 5) |
| Max. SVL females (mm) | 67 (*n* = 9) | 78 (*n* = 15) | 77 | 80 (*n* = 13) | 64 (*n* = 7) | 81 (*n* = 2) | 74 | 74 (*n* = 5) |
| Internasal | Single | Single | Divided | Single | Divided | Single | Divided | Divided |
| Frontal | Single | Single | Single | Single | Single (rarely divided) | Single | Single | Divided |
| Subcaudals per caudal segment | 3 | 4 | 3 | 3 | 3 | 5–6 | 3 | 3 |
| Supralabials | 5–5 | 3–5 | 4–6 | 4–6 | 4–5 | 3–5 | 4–5 | 4–6 |
| Infralabials | 4–4 | 3 | 3–5 | 3–4 | 3–5 | 2–3 | 4–5 | 4–6 |
| Femoral pores | 7–9 | 7–9 | 4–9 | 7–10 | 8–11 | 5–15 | 6–10 | 3–9 |
| Subdigital lamellae on 4th finger | 14–18 | 14–23 | 12–18 | 14–19 | 17 (*n* = 1) | 20–22 | 12–17 | 13–16 |
| Subdigital lamellae on 4th toe | 23–27 | 23–32 | 20–27 | 21–25 | 28 (*n* = 1) | 30–36 | 19–25 | 20–22 |
| Temporal and nuchal scales | A pair of sharp spine like scales in temporal and nuchal region | Low rounded tubercles surrounded by granular scales | Temporal region tuberculated, and nuchal region with two rows of enlarged subconical tubercles on each side of the body | Low tubercles in temporal region and subconical tubercles in nuchal region | Flat temporal scales and low enlarged tubercles in nuchal region | Flat temporal scales and low tubercles surrounded by granular scales in nuchal region | Flat temporal scales and enlarged, slightly keeled nuchal tubercles surrounded by granular scales | Flat temporal scales and enlarged, slightly keeled nuchal tubercles surrounded by small rounded scales |
| Scales of dorsum | Vertebral row of paired, enlarged, keeled scales and two paravertebral series of short oblique rows of projecting scales | Two paravertebral rows of tubercles or spine-like scales, separated by 4–6 small scales | Two paravertebral rows of spine-like scales, separated by 5 or mores irregular scales | Vertebral row of paired, enlarged, smooth scales | Paravertebral rows formed by spine-like scales, discontinuous posteriorly | No continuous vertebral or paravertebral rows of enlarged scales: enlarged scales arranged obliquely in short series of 3–8 scales forming irregular chevrons on dorsal field of body and broken undulating rows of scales along dorsolateral body | Two paravertebral rows separated by 3-8 small irregular scales | Two paravertebral rows of tubercular scales arranged in zigzag pattern |
| Scales of flanks | Enlarged spiny scales forming oblique lines | Tubercular scales forming oblique lines on lateral surface of body | Series of spine-like scales forming oblique lines on lateral surface of body | Series of spine-like scales forming oblique lines on lateral surface of body | Series of conical scales of different sizes forming oblique lines on lateral surface of body | Alternate tubercular scales on lateral surface of body | Series of spine-like scales forming oblique lines on lateral surface of body | Series of tubercular scales forming oblique lines on lateral surface of body |
| Source | This study | *Köhler, Bohme & Schmitz (2004)* | *Vásquez-Restrepo et al. (2020)* | *Uzzell Jr (1965)*; *Köhler, Bohme & Schmitz (2004)*; This study | *Köhler, Bohme & Schmitz (2004)*; *Vásquez-Restrepo et al. (2020)*; This study | *Fritts, Almendariz & Samec (2002)* | *Vásquez-Restrepo et al. (2020)* | *Vásquez-Restrepo et al. (2020)* |

**Notes.**

n, number of specimens studied; SVL, snout–vent length.

Yánez-Muñoz et al. (2021), *PeerJ*, DOI 10.7717/peerj.12523

**Table 2 Morphometric data of the type series of *Echinosaura fischerorum* sp. nov.** Holotype is marked with *. Accompanying the range, the sample size is included in parentheses (N), followed by the mean and standard deviation. All specimens are from the same collection (DHMECN).

| Character | Males | | | | Range | Females | | | | Range | Juveniles | | | Range |
|---|---|---|---|---|---|---|---|---|---|---|---|---|---|---|
| | 15208* | 15209 | 15211 | 16110 | | 14058 | 14060 | 15210 | 16109 | | 14059 | 14061 | 12767 | |
| SVL | 69 | 62.8 | 54.7 | 63.1 | 69–54.7 (4) 66.05 ± 4.17 | 66.8 | 63.8 | 65.2 | 61.4 | 66.8–61.4 (4) 64.1 ± 3.81 | 29.6 | 52.7 | 40.5 | 52.7–29.6 (3) 40.93 ± 11.56 |
| TRL | 33.6 | 27.2 | 28.2 | 28.1 | 33.6–22.7 (4) 30.04 ± 4.52 | 25.5 | 26.1 | 29.5 | 27.6 | 29.5–25.5 (4) 27.5 ± 2.82 | 13.1 | 19.6 | 15.9 | 19.6–13.1 (3) 16.2 ± 3.26 |
| TAL | 89.5 | 85.2 | 67 | 88.2 | 89.5–67 (4) 78.25 ± 15.91 | 82.5 | 84 | 88.7 | 91 | 91–82.5 (4) 86.75 ± 6.01 | 37.5 | 73.6 | 55.6 | 73.6–37.5 (3) 55.57 ± 18.05 |
| SL | 13.7 | 11.8 | 10.8 | 12 | 13.7–10.8 (4) 12.25 ± 2.05 | 12.1 | 12.2 | 12.6 | 12 | 12.6–12 (4) 12.3 ± 0.42 | 5.7 | 9.4 | 7.3 | 9.4–5.7 (3) 7.47 ± 1.86 |
| HL | 20.1 | 17.5 | 17.6 | 18.3 | 20.1–17.5 (4) 18.8 ± 1.83 | 18.5 | 17.2 | 17.2 | 16.3 | 18.5–16.3 (4) 17.4 ± 1,55 | 8.8 | 15.2 | 11 | 15.2–8.8 (3) 11.67 ± 3.25 |
| HW | 13.2 | 12.1 | 10.7 | 11.9 | 13.2–10.7 (4) 11.95 ± 1.76 | 11.6 | 10.7 | 10.6 | 11 | 11.6–10.6 (4) 11.1 ± 0.70 | 5.2 | 9.5 | 7.1 | 9.5–5.2 (3) 7.27 ± 2.15 |
| HUM | 9.7 | 7.9 | 8.4 | 7.9 | 9.7–7.9 (4) 8.8 ± 1.27 | 10.9 | 9.3 | 8.4 | 9 | 10.9–8.4 (4) 9.65 ± 1.76 | 4.6 | 7.7 | 6.4 | 7.7–4.6 (3) 6.23 ± 1.56 |
| FAL | 9.1 | 7.4 | 7.3 | 8.3 | 9.1–7.3 (4) 8.2 ± 1.27 | 8 | 7.7 | 7.1 | 8.1 | 8.1–7.1 (4) 7.6 ± 0.70 | 3.9 | 6.1 | 4.7 | 6.1–3.9 (3) 4.9 ± 1.11 |
| HND | 9.3 | 8.4 | 9 | 8.6 | 9.3–8.4 (4) 8.85 ± 0.63 | 9.7 | 14.8 | 8.2 | 9.9 | 14.8–8.2 (4) 11.5 ± 4.66 | 4.4 | 8.4 | 5.8 | 8.4–4.4 (3) 6.2 ± 2.03 |
| FEM | 12.5 | 11 | 10.7 | 13.1 | 13.1–10.7 (4) 11.9 ± 1.69 | 11.6 | 12.1 | 9.6 | 11.2 | 12.1–9.6 (4) 10.85 ± 1.76 | 5.5 | 9.6 | 6.3 | 9.6–5.5 (3) 7.13 ± 2.17 |
| TIB | 13.7 | 11.4 | 10.9 | 11.3 | 13.7–10.9 (4) 12.3 ± 1.98 | 10.7 | 10.6 | 9.9 | 10.8 | 10.8–9–9 (4) 10.35 ± 0.63 | 5.3 | 9.8 | 5.8 | 9.8–5.3 (3) 6.97 ± 2.47 |
| FTL | 15.7 | 14.7 | 14.9 | 14.2 | 15.7–14.2 (4) 14.95 ± 1.06 | 15.2 | 14.9 | 15.5 | 15.4 | 15.5–14.9 (4) 15.2 ± 0.42 | 7.3 | 13.8 | 8 | 13.8–7.3 (3) 9.70 ± 3.57 |

**Notes.**

SVL, snout–vent length; TRL, trunk length; TAL, tail length; SL, snout length; HL, head length; HW, head width; HUM, humerus length; FAL, forearm length; HND, hand length; FEM, femur length; TIB, tibia length; FTL, foot length.

**Table 3  Lepidotic characters of the type series of *Echinosaura fischerorum* sp. nov.**

| Character | Males | Females | Juveniles |
|---|---|---|---|
| Frontal scales | 1 | 1 | 1 |
| Large chin shields | 2 | 2 | 2 |
| Gulars in transverse row | 19–20 | 17–21 | 18–19 |
| Internasal scales | 1 | 1 | 1 |
| Frontonasal scales | 2 | 2 | 2 |
| Frontoparietal scales | 2 | 2 | 2 |
| Supraoculars | 3 | 3 | 3 |
| Presupraoculars | 1 | 1 | 1 |
| Postmentals | 1 | 1 | 1 |
| Mentals | 1 | 1 | 1 |
| Scales between paravertebral edges | 10–11 | 10–11 | 9–10 |
| Supralabial scales | 5–5 | 5–5 | 5–5 |
| Infralabial scales | 4–4 | 4–4 | 4–4 |
| Femoral pores | 8–9 | 0 | 7–8 |
| Lamellae under the fourth finger: left, right | 16–18/14–18 | 16–18/15–17 | 15–18/14–17 |
| Lamellae under the fourth toe: left, right | 24–26 | 24–27/24–26 | 24–25/23–26 |
| Subcaudal scales per autotomic segment | 3 | 3 | 3 |

(DHMECN 14061) or a broad vertebral stripe along dorsum (DHMECN 12767, 14059), with a conspicuous dorsal transverse bar on base of tail. Regardless of sex or age, the snout (rostral, internasal and prefrontal scales) varies between cream, orange cream, and dark brown; gular background varies from dark brown to reddish brown; coloration of the belly can be predominantly dark brown. In life, the dorsal pale spots vary from yellowish cream to orange cream (Fig. 6).

**Hemipeneis of holotype (Fig. 7).** The lobes of the expanded, heart-shaped right hemipenis, are not fully everted; it is 9.2 mm long and 8.7 mm wide. It has a globular hemipenial body, thinner at the base and broader distally, and bears two small symmetrical lobes. The sulcus spermaticus is defined by two thick parallel margins and extends medially in a straight line along the hemipenial body up to the lobular crotch, where a fleshy fold divides it into two branches that run medially on each lobe. Five short, oblique, and parallel flounces bearing five to seven calcified spicules each lie on each side of the sulcus spermaticus with a narrow nude area in between. Lateral and asulcate aspects of hemipenis are ornamented distally with eight approximately equidistant flounces bearing rows of calcified spicules; of these, the five distal flounces converge on the asulcate face forming chevrons with apices directed toward base of hemipenis, while the other three flounces increasingly diverge toward base of organ. On the asulcate side, these flounces are separated medially by a large naked region. Ventral to the flounces, the lateral and asulcate sides of the hemipenial body are covered by 41 large, evenly distributed spines that increase slightly in size towards the base of the organ.

**Cranial Osteology (Figs. 8, 9, 10).** The following description of the skull of *Echinosaura fischerorum* sp. nov. is based on female paratype DHMECN 15210. The skull is robust,

moderately compressed (skull width = 60% of skull length) and moderately high (skull height = 43% of skull length). The widest region of the skull is at the level of the contact between squamosal and jugal. The dorsal margin of the snout in lateral view is slightly convex. The surface of frontal and parietal is very rugose. Basioccipital, compound otooccipitals (fused opisthotic and exoccipital), prootics and supraoccipital are almost indistinguishably fused and the parabasisphenoid is partly fused to this structure. The mandibles are very robust, and coronoid, splenial and angular are largely fused with the compound bone. Intraspecific variation in cranial osteology of *Echinosaura fischerorum* sp. nov. is presented in Figs. S1 and S2.

## Dermatocranium

**Premaxilla (Figs. 8, 9A).** The premaxilla bears 12 tooth loci. The ascending nasal process of the premaxilla contacts the nasals posteriorly and forms the anterior and the anterodorsal edge of the naris laterally. There is a small foramen anterolaterally on both sides of the premaxilla. The palatal shelf of the premaxilla is notched medially and contacts the premaxillary process of the maxilla.

**Maxilla (Figs. 8, 9A).** The maxilla bears 18 tooth loci, arranged in a straight row. The bone is large and occupies most of the lateral surface of the skull anterior and ventral to the orbit. The anterolateral process of the maxilla is slightly notched anteriorly and articulates with the maxillary process of the premaxilla. The facial process of the maxilla is triangular and forms the posterior edge of the naris. It contacts the nasal and the frontal dorsally, and the prefrontal posterodorsally, extensively overlapping the anterior process of the prefrontal. There is a row of five foramina on the supralabial edge of the facial process. The posterolateral process of the maxilla is posteriorly bifurcate and articulates with the jugal. The anteromedial process of the maxilla is elongate, triangular, and contacts the palatal shelf of the premaxilla laterally and the vomer medially, which represents the only contact point between the maxilla and the vomer. The medial edge of the palatal shelf of the maxilla is irregularly shaped and broadest at the level of maxillary teeth 7–10, tapering abruptly posterior to tooth 10 and ending at the level of tooth 14. It forms a palatine facet, where the palatine articulates at the level of maxillary teeth 11–13. The posteromedial process of the maxilla articulates with the maxillary process of the ectopterygoid.

**Nasal (Figs. 8A, 8C, 9A).** The nasal is ellipsoid in dorsal view, much longer than wide and about twice as long as the premaxilla. Along the anterior half of the nasal its lateral edge articulates with the dorsal edge of the naris and along its posterior half it contacts the facial process of the maxilla laterally. Anteriorly the nasal is convexly shaped and in broad contact with the premaxilla. Both nasals are in broad contact medially. Posteriorly the nasal is in broad contact with the frontal.

**Frontal (Figs. 8, 9).** The anterior margin of the frontal is less than half the width of its posterior edge. The interorbital constriction is distinctly narrower than the anterior edge and at its narrowest point about one fourth the width of the posterior edge. The anterior margin has three pointed processes in dorsal view, of which the lateral two are slightly longer than the medial process. The nasal facets of the frontal are between the medial process and the lateral processes. The lateral process of the anterior margin contacts the

dorsal edge of the facial process of the maxilla. The lateral margin of the frontal contacts the posterodorsal process of the prefrontal anteriorly and the anterior half of the dorsal edge of the postfrontal posteriorly. The median part of the frontal (posterior to the contact zone with the prefrontal and anterior to the contact zone with the postfrontal) is part of the dorsal margin of the orbit. The posterior edge of the frontal has a pair of short frontoparietal tabs (*Roscito & Rodrigues, 2010*) that project slightly and articulate with the frontal facets of the parietal. The cristae cranii originate on the ventrolateral surface of the frontal and meet each other at midline, where they are fused into a tubular structure that extends posteriorly almost to the level where the anterior tip of the postfrontal contacts the frontal. This structure encapsulates the olfactory tract and reaches anteriorly to the level of the nasals. The anterolateral border of the cristae cranii contacts the medial edge of the prefrontal.

**Parietal** (Figs. 8, 9). The parietal is about as long as wide. It is distinctly shorter than the frontal and its widest part is narrower than the posterior end of the frontal. The anterolateral and the posterolateral processes of the parietal taper distally and are oriented outwards. The anterolateral process extends into the suture between the frontal and the postfrontal, which is the only contact zone between the parietal and postfrontal. The posterolateral process contacts the supratemporal and the squamosal posterolaterally, and the paroccipital process of the otoccipital posteriorly. The posterior edge of the parietal bears a posteromedial notch. The lateral margin of the parietal posterior to the anterolateral process forms the dorsal margin of the supratemporal fenestra. A slender post-temporal fenestra is present, delimited anteriorly by the posterior margin of the parietal, and posteriorly by the supraoccipital, separated at the midline by a short anterior projection of the supraoccipital. On the ventral side of the parietal, a triangular descending process originates anterolaterally just posterior to the anterolateral process; the triangular process lies medial to the epipterygoid and reaches its dorsal tip without contact. The descending process lies slighty anterior to the alar process of the prootic.

**Prefrontal** (Figs. 8A, 8C). The prefrontal forms the anterior and anterodorsal margins of the orbit. The orbitonasal flange of the prefrontal contacts medially the anterolateral surface of the cristae cranii of the frontal and the anterior border of the palatine. A lacrimal foramen is present between the internal wall of the facial process of the maxilla and the ventral process of the prefrontal that rests half on the maxillary palatal shelf and half of the palatine. The ventral process of the prefrontal is broadly separated from the anterior process of the jugal. The posterodorsal process of the prefrontal is placed beneath the lateral margin of the frontal and is, thus, not visible in dorsal view.

**Postfrontal** (Figs. 8, 9). The postfrontal is triradiate and forms part of the posterodorsal margin of the orbit, contacting dorsally the posterior part of the frontal and the anterior part of the parietal. The ventral process of the postfrontal contacts the jugal ventrally, forming the postorbital bar and excluding the postorbital from the orbit. Laterally, the postfrontal contacts the postorbital. The posterodorsal process of the postfrontal comprises more than half the total length of the postfrontal and expands posteriorly to form part of the anterior border of the supratemporal fenestra. The acute anterior process of the

postfrontal is triangular, projecting anteromedially, and corresponds to two-thirds of the length of the posterodorsal process.

**Postorbital** (Figs. 8, 9). The postorbital is long and lozenge-shaped. Anteriorly and along the anterior half of its dorsal margin it contacts the postfrontal. The posterior half of its dorsal margin forms the lateral border of the supratemporal fenestra. It contacts the dorsal surface of the anterior half of the squamosal posteriorly.

**Squamosal** (Figs. 8, 9). The squamosal is long and has a hockey-stick shape. The anterior two-thirds are straight and taper into an acute tip, contacting the postorbital dorsally and the posterodorsal ending of the jugal laterally. The posterior part curves downwards to meet the supratemporal medially. The posteroventral tip fits into the deep notch of the tympanic crest of the quadrate. The squamosal forms the posterolateral border of the supratemporal fenestra and the posterodorsal border of the infratemporal fenestra.

**Epipterygoid** (Figs. 8C, 9A). The epipterygoid is a rod-like slender bone with slightly expanded epiphyses. The ventral epiphysis inserts into the fossa columellae of the pterygoid, and the dorsal epiphysis is located anterior to the alar process of the prootic and lateral to the descending process of the parietal, but not contacting these elements.

**Supratemporal** (Figs. 8A, 8C, 9B). The supratemporal is a small, rod-like slender bone which contacts the posterolateral process of the parietal dorsally, the squamosal laterally, the paroccipital process of the otoccipital medially, and the quadrate ventrally.

**Quadrate** (Figs. 8, 9). The quadrate is robust and conchal-shaped. It can be divided into a ventrally located mandibular condyle, articulating with the compound bone of the mandible, and a dorsally located cephalic condyle articulating with the supratemporal, the paroccipital process of the prootic and the ventral tip of the squamosal, which fits into a deep notch in the dorsal surface of the quadrate. A lateral concave tympanic crest, a posterior crest and a medial crest extend between the mandibular and cephalic condyles.

**Jugal** (Figs. 8, 9). The jugal is sigmoidal in shape. It forms part of the ventral and the posterior margins of the orbit. Anteriorly it is bifurcated with a longer pointed suborbital process and a shorter more rounded maxillary process. The robust dorsal temporal process contacts the ventral margin of the postfrontal, the postorbital and the anterior process of the squamosal, where it ends in a blunt tip. Medially, the jugal has a triangular ectopterygoid process, which contacts the lateral margin of the ectopterygoid. On its lower edge the jugal has a long and robust triangular free posterior process.

**Vomer** (Fig. 8B). The elongate vomer is the most anterior component of the skull floor located posterior to the premaxillary palatal shelf and attached to a cavity in it. The anterior process tapers to a pointed tip. The lateral shelf of the vomer has an irregular border that partly overlaps the palatal shelf of the maxilla dorsally. There is an anterolateral process that overlaps the vomeronasal fenestrae without contacting the maxillary palatal shelf. Posteriorly, the slender palatine process of the vomer overlaps ventrally the vomerine process of the palatine. The dorsal surface of the vomer has a curved transversal crest that contacts the septomaxilla. Posterior to the crest is a large vomerine foramen. There is a dorsal crest along the medial border of the vomer, which contacts its counterpart on the other vomer.

**Palatine** (Fig. 8). The palatine forms the middle portion of the skull floor. It is anteriorly wider and narrows posteriorly to meet the pterygoid. Anteromedially, the vomerine process overlaps the posterior end of the vomer. Anterolaterally, the wide maxillary process of the palatine articulates with the maxillary palatal shelf. Between the medial and lateral extremities of its anterior margin the palatine has a ventral concave surface, deeper anteriorly and gradually shallower posteriorly, which forms the choanal channel. Posteriorly, the pterygoid process overlaps dorsally the medial palatine process of the pterygoid. The lateral margin of the posterior part of the palatine forms the medial margin of the suborbital fenestra.

**Pterygoid** (Figs. 8, 9). The Y-shaped pterygoid forms the posterior part of the skull floor. Anteriorly, the pterygoid is divided into a medial lanceolate palatine process and a lateral triangular and broad ectopterygoid process, forming the posterior margin of the suborbital fenestra. The palatine process overlaps with the pterygoid process of the palatine. The ectopterygoid process fits into a facet between the dorsal and ventral portions of the posteromedial process of the ectopterygoid. Posteriorly, the long and slender quadrate process extends dorsolaterally to almost reach, but not contact the medial crest of the quadrate. The basipterygoid process of the parabasisphenoid contacts the pterygoid medially at about midlength. The dorsomedial surface of the pterygoid has a shallow groove, the fossa columellae, into which the ventral epiphysis of the epipterygoid inserts.

**Ectopterygoid** (Figs. 8B, 8C). The ectopterygoid is a short but robust bone that forms part of the posterior and external margin of the suborbital fenestra. Anteriorly, its maxillary process articulates with the posteromedial process of the maxilla. Laterally, it contacts the triangular ectopterygoid process of the jugal. Posteriorly, the posteromedial process of the ectopterygoid articulates with the ectopterygoid process of the pterygoid.

**Septomaxilla** (Fig. 8C). The dome-shaped septomaxilla is located between the vomer and the premaxilla, and rests laterally on the palatal shelf of the maxilla. Dorsally it reaches close to the ventral side of the nasal process of the premaxilla.

## Neurocranium

**Parabasisphenoid** (Figs. 8B, 8C, 9B). The body of the parabasisphenoid is roughly rectangular and bears two long basipterygoid processes that project anterolaterally to meet the pterygoid medially. These processes are flattened near the facet of the pterygoid. Anteriorly, between the basipterygoid processes is a rectangular parasphenoid rostrum. The parabasisphenoid is partly fused to the prootic posterolaterally and to the basioccipital posteriorly.

**Prootic** (Figs. 8C, 9A). The prootic forms the anterior portion of the otic capsule. It has an anteriorly directed curved and compressed alar process, which approaches the descending process of the parietal and the epipterygoid but does not contact those structures. Below the alar process is the C-shaped incisura prootica, a deep and round notch and exit for the trigeminal nerve. The posterior border of the prootic is marked by the fenestra ovalis, into which the rounded footplate of the stapes inserts. The crista prootica begins at the parabasisphenoid and runs across the lateral side of the prootic. Posterior to its tip and at about the same level as the trigeminal notch is the small foramen for the facial

nerve. Dorsally, the prootic contacts the supraoccipital. Ventrally, it articulates with the parabasisphenoid anteromedially and with the basioccipital posteromedially.

**Otooccipital** (**Figs. 8A**, **8C**, **9B**). The compound otooccipital (fused opisthotic and exoccipital) forms the posterior portion of the otic capsule. It contributes to the composition of the occipital condyle and forms the lateral margin of the foramen magnum. The otooccipital contacts the supraoccipital dorsally and the basioccipital ventrally. Its anterior border contacts the prootic, and has a C-shaped notch, which forms the posterior margin of the fenestra ovalis, and its anterodorsal margin is thickened to form the paroccipital process. Posterolaterally to the fenestra ovalis is the paroccipital process, which contacts the supratemporal and the quadrate laterally. The sphenoccipital tubercle is situated at the junction of the otooccipital and the basioccipital. This tubercle is ventral to the fenestra ovalis, and between both structures is an ellipsoid lateral foramen.

**Basioccipital** (**Figs. 8A**, **9B**). The pentagonal-shaped basioccipital forms most of the braincase floor. Anteriorly, it is fused to the parabasisphenoid, dorsally it contacts the prootic (anterior) and the otooccipital (posterior). It forms the ventral margin of the occipital condyle.

**Supraoccipital** (**Figs. 8A**, **9B**). The hourglass-shaped broad supraoccipital forms the roof of both the braincase and otic capsules. The short ascending process of the tectum synoticum originating at the anterior margin of the supraoccipital does not reach the parietal bone. A large posttemporal fenestra is present between the supratemporal and the parietal. The supraoccipital articulates with the dorsal margin of the prootic anterolaterally and with the otooccipital posterolaterally. The posterior border forms the dorsal edge of the foramen magnum.

**Orbitosphenoid** (**Figs. 8B**, **8C**). The small paired orbitosphenoids are compressed and L-shaped with the vertex oriented laterally and bearing a small lateral process. The dorsal process is broad and both bones are completely fused medially along this process, forming a slightly bifurcate anterior end of the fused bone. On the ventral surface of the fused dorsal region is a short median, comparatively broad anteroventrally oriented process. The ventral processes of the fused orbitosphenoid bone are slender, directed medially, but distinctly separated. The orbitosphenoid is located anterior to the braincase and posterior and medially with respect to the orbits, at an angle to the dorsoventral axis of the skull.

**Stapes** (**Figs. 8B**, **9A**). The ossified stapes is composed of a long and slender laterally projecting shaft and a large, rounded footplate, which entire fills the area of the fenestra ovalis. The shaft ends proximal to the posterior crest of the quadrate.

### Mandible

**Dentary** (**Figs. 9**, **10**). The dentary is the anterior part of the mandible and bears a straight row of 22 (right) to 23 (left) teeth loci. Posteriorly, the dentary contacts the coronoid, angular, splenial and surangular. The labial and anteromedial processes of the coronoid clasp the posterodorsal surface of the dentary. Anteriorly, at the lingual surface of the dentary, the opening of Meckel's canal is observed near the mandibular symphysis. The dentary has about five foramina, the anteriormost two are located on the symphyseal region, and the posteriormost is at the level of the midpoint of the dental row.

**Coronoid** (Figs. 9A, 10). The triradiate coronoid has a long and robust free dorsal coronoid process. The labial process tapers anteriorly up to the level of the gap between the third and fourth posteriormost teeth of the dentary. On the lingual surface of the mandible, the anteromedial process of the coronoid contacts the dentary anteriorly, the splenial ventrally, and the surangular posteriorly. The slender posteromedial process contacts the surangular, but the exact borders are not well resolved in the CT-images. There is a small medial crest between the lower part of the coronoid process and the posteromedial process.

**Angular** (Figs. 9B, 10A, 10B, 1D). The small and slender angular has a pointed anterior and a blunt posterior end. It is located on the ventral surface of the mandible, ventral to the angular process of the dentary. It contacts the surangular posteriorly and the ventral border of the splenial medially. The posterior mylohyoid foramen is visible ventrally at the posterior end where the angular meets the surangular.

**Splenial** (Figs. 9B, 10A, 10B, 10D). The splenial forms the midventral portion of the lingual surface of the mandible. It contacts the dentary anteriorly, the coronoid dorsally, the angular ventrally, and the compound bone posterodorsally. There are two foramina present in the anterior region of the splenial: a larger oval anterior inferior foramen at the suture with the dentary, and a more rounded anterior mylohyoid foramen about half as large, located ventral and posterior to the anterior inferior foramen. The exact borders of the splenial are not well resolved in the CT-images, making it difficult to comment about the exact shape of the bone.

**Mandibular Compound Bone** (Figs. 9, 10). The compound bone is the posterior part of the mandible and is formed by the fusion of the surangular on the labial surface, and the prearticular and articular on the lingual surface of the mandible. The prearticular forms the floor and the internal margin of the adductor fossa, while the articular forms its posterior margin and provides the articulation to the skull on its dorsal surface. The surangular is located posterior to the dentary and the coronoid and a large surangular foramen is observed next to the contact with the coronoid. Ventral to the surangular is the angular. The retroarticular process is formed by the fused surangular and prearticular-articular complex.

**Vertebrae** (Figs. 15, 16). There are two sacral vertebrae and 25 presacral vertebrae of which the first three and the last one do not bear ribs.

**Pectoral Girdle** (Figs. 15, 16). The pectoral girdle consists of a clavicle, suprascapular, interclavicle, fused scapula and coracoid, epicoracoid, sternum, and xiphisternum. The clavicle is large, triangular, flattened, and encloses a large fenestra. The anterior part of the interclavicle is trifurcate with one median and two anterolaterally oriented processes, each of them about one-fourth the total length of the interclavicle and tapering anteriorly. The median process penetrates anteriorly the posterior third of the suture between the two clavicles. Posteriorly, it has a long, straight, rod-like and posteriorly tapering branch, which surpasses the central part of the sternal fontanelle. The large scapulocoracoid bears a small coracoid fenestra, a large anterior coracoid fenestra, a medium-sized posterior coracoid fenestra, a large scapulocoracoid fenestra, and the glenoid fossa. The rhomboid sternum has a large central fontanelle and is associated with three pairs of sternal ribs. The rod-like xiphisternum has two fenestrae and receives two pairs of xiphisternal ribs.

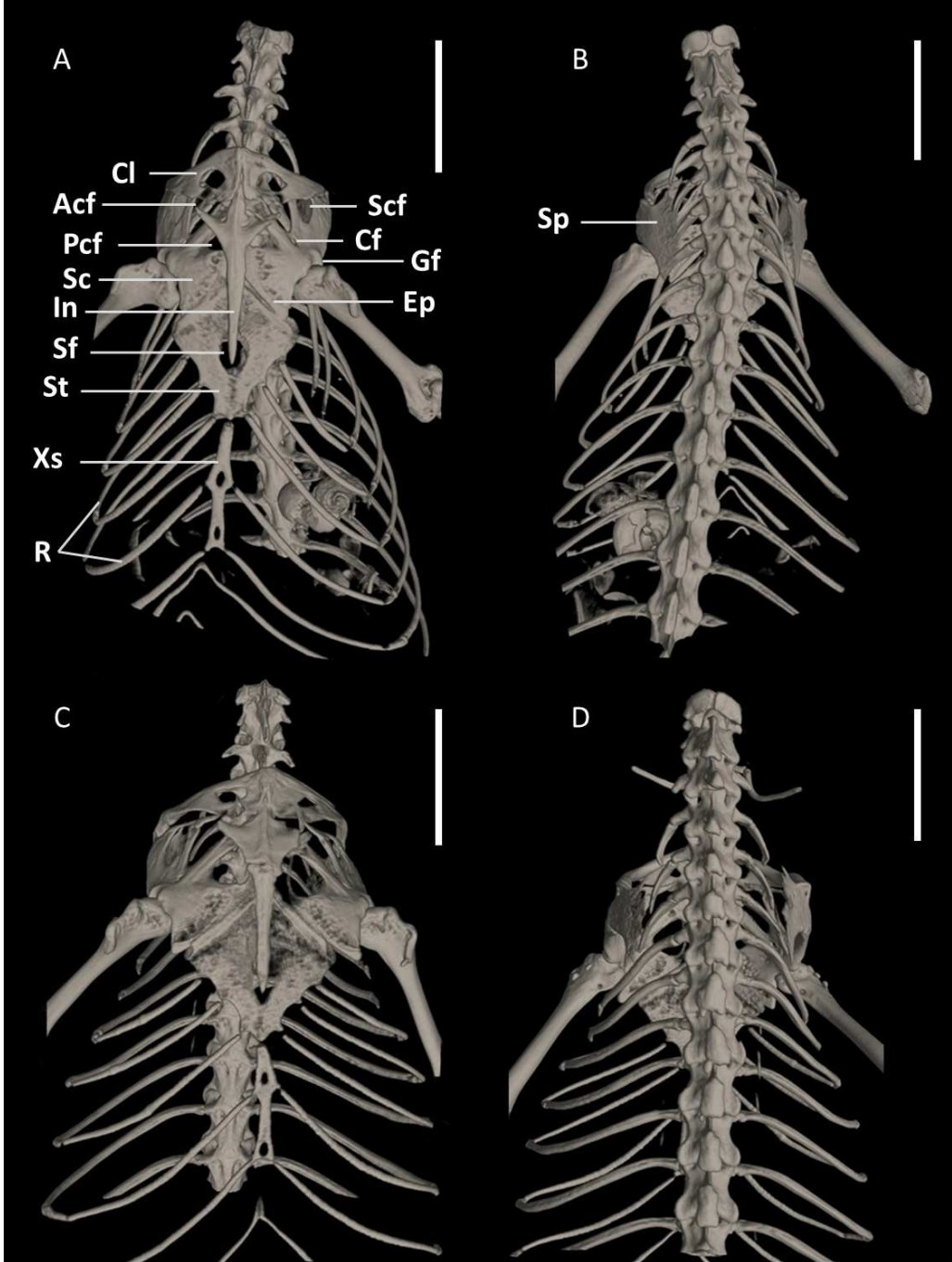

**Figure 15** **Pectoral girdles of female paratype DHMECN 15210 (A, B) and male holotype DHMECN 15208 (C, D) of *Echinosaura fischerorum* sp. nov. in ventral (left) and dorsal (right) views.** Acf, anterior coracoid fenestra; Cf, coracoid fenestra; Cl, clavicle; Ep, epicoracoid; Gf, glenoid fossa; In, interclavicle; Pcf, posterior coracoid fenestra; R, ribs; Sc, scapulocoracoid; Scf, scapula-coracoid fenestra; Sf, sternal fenestra; Sp, suprascapula; St, Sternum; Xs, xiphisternum. Scale bars = 5 mm.

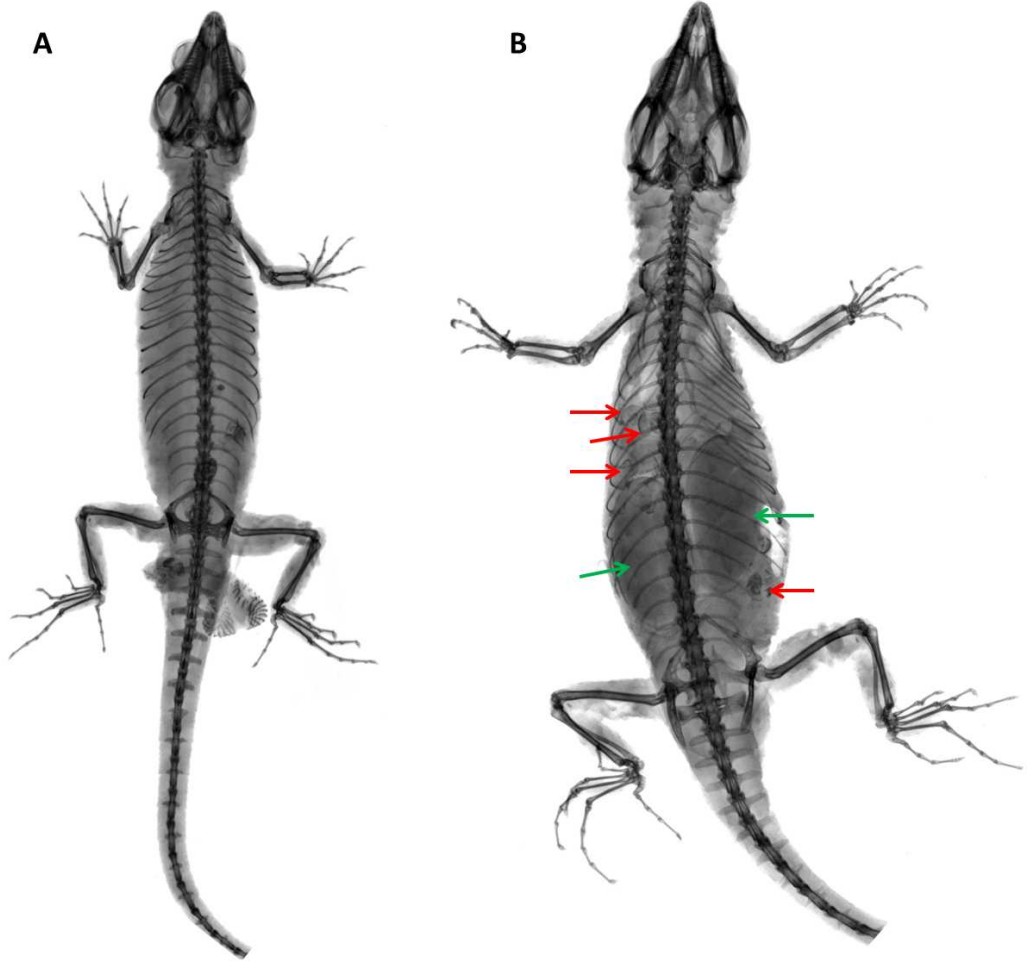

**Figure 16  X-ray images of *Echinosaura fischerorum* sp. nov. (A) Male holotype (DHMECN 15208); (B) female paratype (DHMECN 15210).** Red arrows show snail shells in the stomach; green arrows show eggs.

The suprascapula is not visible in ventral view and contacts the scapulocoracoid and the clavicle laterally. The epicoracoid contacts the median edge of the coracoid, but most of this structure is overlain by the interclavicle.

**Distribution and Natural History (Figs. 16, 17).** *Echinosaura fischerorum* sp. nov. is known from the western slopes of the Andes in northwestern Ecuador, Imbabura and Carchi Provinces, between 1,495–1,750 m (Fig. 17). All known specimen records lie in the lower montane forest ecosystem of the Mira river basin within the Dracula Reserve, a 1,136 ha private protected area managed by the Ecominga Foundation. The record from Imbabura province corresponds to a photograph by Jaime Culebras from Manduriacu Reserve, also managed by the Ecominga Foundation. Most known localities of *E. fischerorum* sp. nov. are in close proximity to the Ecuador-Colombia border and we expect that this species might also be present in neighboring Colombia. Most specimens were observed active among tree roots during the day and collected in pitfall traps along forest ridges near streams.
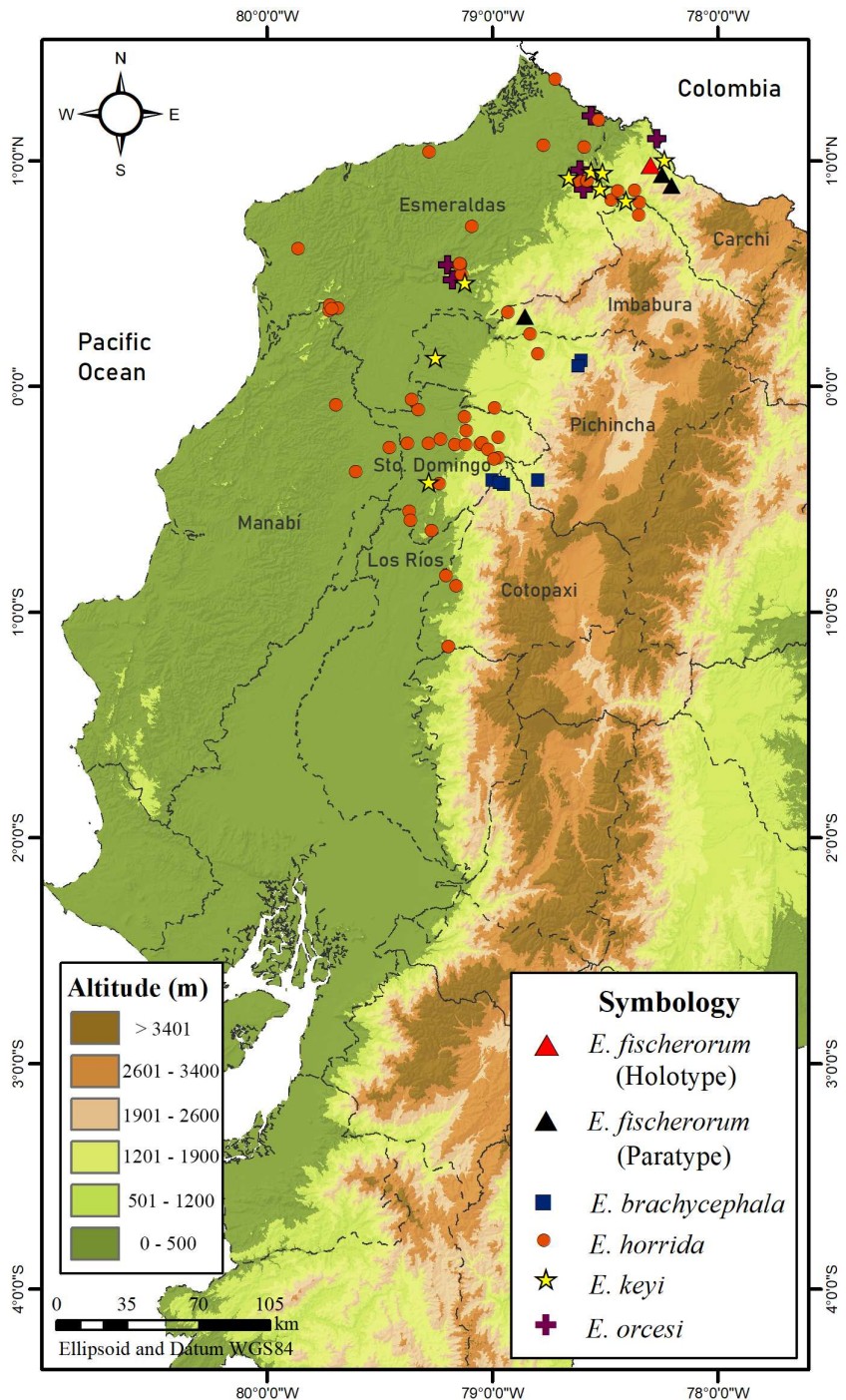

**Figure 17** Map of distribution of species of *Echinosaura* in Ecuador.

Gravid females were collected in April and November. One female paratype (DHMECN 15210) contained two large oval eggs in the oviduct and several snail shells in the stomach (Fig. 16).

**Etymology.** The specific epithet is a patronym in honor of Beat Fischer and Urs Fischer, donors who have contributed significantly to the consolidation of the Dracula Reserve in the sectors of Peñas Blancas and El Pailón, which not only protect the populations of this new endemic species, but also important populations of threatened amphibians and reptiles of the Mira river basin.

## DISCUSSION

*Vásquez-Restrepo et al. (2020)* presented an extensive review of *Echinosaura* based on molecular phylogeny, external morphology and biogeography. However, no material was available to those authors for review from extreme southwestern Colombia and northwestern Ecuador, that would have offered the possibility of recognizing additional candidate species of *Echinosaura*.

The new species is genetically distinct from its congeners and is readily distinguishable based on external morphological characters. By contrast, information on internal morphology of *Echinosaura* is very scarce and to date only few studies (*Bell, Evans & Maisano, 2003*; *Estes, de Queiroz & Gauthier, 1988*; *Evans, 2008*; *MacLean, 1974*) have been published with exemplary information on cranial morphology of *Echinosaura*. Furthermore, this information is based only on one specimen of *E. panamensis* (CAS 99994) and two specimens of *E. horrida* (MCZ 18858, USNM 163424) (*Bell, Evans & Maisano, 2003*; *Evans, 2008*). Here, for the first time a detailed description of the cranial morphology of *Echinosaura* is provided based on CT data of a paratype (DHMECN 15210) of the new species herein described. Unlike *E. horrida* and *E. panamensis*, the nasal bones of *E. fischerorum* sp. nov. are not separated and thus the premaxilla does not meet the frontal. The postfrontal is not excluded from the upper temporal fenestra as indicated by *Evans (2008)* for *E. horrida*, but rather expands posteriorly to form the anterior border of the fenestra. Although information on skull morphology of *Echinosaura* is scarce, our data show remarkable interspecific variation suggesting that skull morphological characteristics have taxonomic value. Future studies should therefore examine and compare the skulls of other species of *Echinosaura* to obtain a better overall picture of the skull of this taxon. In addition, to estimate the degree of intraspecific variation and thus avoid misinterpretation at the species level, several individuals per species should be examined. Due to possible cryptic diversity within widespread linages, it is also advisable to include type material in the analyses, if possible, in order to be able to make reliable statements about the cranial osteology of the individual species.

The recognition of *Echinosaura fischerorum* sp. nov. improved our understanding of the diversity and evolutionary history of *Echinosaura*. However, we acknowledge that further phylogenetic and phylogeographic analyses of the populations assigned to *Echinosaura fischerorum* sp. nov. from the Río Manduriacu Reserve are needed.

The new species described in this paper occurs in a particular area of high endemism and biological diversification in extreme northwestern Ecuador. Therefore, we expect its

occurrence in the neighboring country of Colombia, a distribution pattern observed in other species of small vertebrates (*Arteaga et al., 2016*; *Yánez-Muñoz et al., 2020*; *Brito et al., 2020*; *Reyes-Puig et al., 2020*).

In terms of conservation, although the type locality of *Echinosaura fischerorum* sp. nov. is part of the ecological reserves managed by the Ecominga Foundation, the entire area is threatened by human impacts such as mining concessions and land use changes for logging. The IUCN conservation status of this new species should be evaluated as soon as possible in further studies, as it is very likely to be threatened.

## CONCLUSIONS

Fieldwork in the western foothills of the Andes in northwestern Ecuador led us to the discovery of a new species of *Echinosaura* and increased the number of known species of this genus to eight.

Our phylogenetic analyses support the monophyly of *Echinosaura fischerorum* sp. nov. While its sister species *E. horrida* is relatively widespread in the tropical lowland rainforest of western Ecuador between sea level and ∼1,500 m, the record of *E. fischerorum* sp nov. from Manduriacu suggests that this species is most likely widespread between Manduriacu Reserve and Río Mira at slightly higher elevations.

## ACKNOWLEDGEMENTS

We want to express our gratitude to the Rainforest Trust and their "Species Legacy" program, to Orchid Conservation Alliance, and University of Basel for their support of Ecominga Foundation's efforts to protect the forest where the new species was discovered. Special thanks to Javier Robayo, Lou Jost, Beat Fischer and Urs Fischer for their continuous support since the beginning of Dracula Reserve. We appreciate the institutional support given by Diego Inclan and Francisco Prieto of INABIO and we are greatful to the support of Miriam Factos of GIZ Ecuador. We thank the collaboration during field work by Callie Broadus, Jaqui Curay, Rocío Manovandas, Rubí García, Jorge Brito, Glenda Pozo, Jordi Salazar, Fausto Recalde, Mateo A. Vega-Yánez, Gabriela Puetate, Pearson McGovern, Natalia Espinoza, Daniel Valencia, Marco Montero, Andy Better, Julio Carrión, Mauricio Herrera-Madrid, Roberto Taicus and the "Dracula's Rangers" of Ecominga Foundation: Hector Yela, Milton Canticuz, Jeovany Guerra, Rolando Peña, Nilo Ortiz, and David Yela. Ministerio de Ambiente de Ecuador issued research permits and framework agreement for access to genetic resources. We are grateful to Patrick Campbell (BMNH), Andreas Schmitz (MHNG), Silke Schweiger and Georg Gassner (NMW) for loan of specimens and to Morris Flecks (ZFMK) for taking photographs. Chris Phillips (UIMNH) kindly sent us photographs of the holo- and paratype of *E. keyi*.

## APPENDIX. SPECIMENS EXAMINED

\* Type specimens

*Echinosaura fischerorum* sp. nov. ($n = 9$): ECUADOR: Carchi: Reserva Dracula, Sector El Guapilal: DHMECN 15208*, DHMECN 15209*, DHMECN 15210*, DHMECN 15211*; Reserva Dracula, Sector El Pailón Chico: DHMECN 14058*, DHMECN 14059*, DHMECN 14060*, DHMECN 14061*; Reserva Dracula, Cerro Oscuro: DHMECN 12767*.

*Echinosaura brachycephala* ($n = 7$): ECUADOR: Cotopaxi: Las Pampas (= San Francisco de las Pampas, −0.418, −78.966, 1,275 m elevation): MHNG 2359.39, ZFMK 46370*, ZFMK 46371*, ZFMK 46372*; Pichincha: Santa Lucía Cloudforest Ecological Reserve (0.0922, −78.620, 1,800 m): QCAZ 11913, QCAZ 11918; Santa Lucía Cloudforest Ecological Reserve, Gallo de la Peña trail (0.117, −78.607, 1,932 m): QCAZ 10824.

*Echinosaura horrida* ($n = 6$): ECUADOR: ZFMK 7272; Pacific versant of Ecuador: ZFMK 43762, ZFMK 43763, ZFMK 46369; Esmeraldas: Tesoro Escondido Reserve (0.4969600, −79.13682, 620 m): QCAZ 15030; Manabí: Zapote (−0.375, −79.605, 154 m): QCAZ 14506.

*Echinosaura keyi* ($n = 2$): ECUADOR: ZFMK 76379; Pichincha: E Rio Baba bridge, 24 km S Santo Domingo de los Colorados, 600 m: UIMNH 80541*.

*Echinosaura orcesi* ($n = 4$): ECUADOR: Esmeraldas: road Alto Tambo - El Placer (0.9, −78.616, 585 m): QCAZ 6299; Tesoro Escondido Reserve, Camarón river (0.544, −79.142, 243 m): QCAZ 15026; Carchi: San Marcos, 670 m: NMW 32000:1*; San Marcos, 700 m: NMW 32000:2*.

*Echinosaura palmeri* ($n = 1$): COLOMBIA: Chocó: Noananoá, Río San Juan, 30 m: BMNH 1923.10.12.14.

*Echinosaura panamensis* ($n = 7$): PANAMA: Coclé: 5–6 km N El Copé: ZFMK 45779, ZFMK 50084, ZFMK 50085, ZFMK 50462, ZFMK 50464, ZFMK 52200, ZFMK 54631.

### Funding

The "Germany-Brazil-Ecuador Trilateral Cooperation Program" of the GIZ international cooperation provided financial support to Mario H. Yánez-Muñoz and Claudia Koch. Omar Torres-Carvajal was financially supported by the Deutscher Akademischer Austauschdienst (DAAD). Laboratory work was funded by a grant from SENESCYT (Arca de Noé Initiative; S. R. Ron and Omar Torres-Carvajal principal investigators). The work of Mario H. Yánez-Muñoz and Juan P Reyes-Puig is part of the research program Diversidad de Pequeños Vertebrados de Ecuador, supported by INABIO through the project: "Conservación de la Biodiversidad en la Cuenca Binacional de los Ríos Mira-Mataje (MMRB): Construcción de Bases Biofísicas y Socio-ambientales para la Conservación y el Manejo Adaptativo de Servicios Ecosistémicos". The funders had no role in study design, data collection and analysis, decision to publish, or preparation of the manuscript.

### Grant Disclosures

The following grant information was disclosed by the authors:
The GIZ international cooperation.
The Deutscher Akademischer Austauschdienst (DAAD).

SENESCYT (Arca de Noé Initiative; S. R. Ron and Omar Torres-Carvajal principal investigators).
INABIO.

## Competing Interests

The authors declare there are no competing interests.

## Author Contributions

- Mario H. Yánez-Muñoz, Omar Torres-Carvajal, Juan P. Reyes-Puig and Claudia Koch conceived and designed the experiments, performed the experiments, analyzed the data, prepared figures and/or tables, authored or reviewed drafts of the paper, and approved the final draft.
- Miguel A. Urgiles-Merchán performed the experiments, analyzed the data, prepared figures and/or tables, authored or reviewed drafts of the paper, and approved the final draft.

## Animal Ethics

The following information was supplied relating to ethical approvals (i.e., approving body and any reference numbers):

We conducted this study under research permits MAE-DNB-CM-2016-0045 and MAE-DNB-CM-2019-0120, issued by the Ministerio del Ambiente del Ecuador. We followed the guidelines for use of live amphibians and reptiles in field research Beaupre et al., 2004, compiled by the American Society of Ichthyologists and Herpetologists, the Herpetologists' League and the Society for the Study of Amphibians and Reptiles.

## Field Study Permissions

The following information was supplied relating to field study approvals (i.e., approving body and any reference numbers):

Ministerio de Ambiente de Ecuador issued research permits and framework agreement for access to genetic resources (MAE-DNB-CM-2016-0045 and MAE-DNB-CM-2019-0120).

## DNA Deposition

The following information was supplied regarding the deposition of DNA sequences:

The DNA sequences of the genes 12S, 16S, ND4, and c-mos are available via GenBank accession numbers MW525208–09, MW525211–12, MW512698–701.

## Data Availability

The CT-Data set is available at MorphoSource (media ID 000381846, DOI 10.17602/M2/M381846; Project 4120, DOI: 10.7934/P4120).

## New Species Registration

The following information was supplied regarding the registration of a newly described species:

Publication LSID: urn:lsid:zoobank.org:pub:9D513A0D-D676-4290-AE5D-A128B7DBE3F3

Echinosaura fischerorum sp. nov.: urn:lsid:zoobank.org:act:F8DCFE99-4862-4476-9A5C-EB684D0CD73A.

## Supplemental Information

Supplemental information for this article can be found online at http://dx.doi.org/10.7717/peerj.12523#supplemental-information.

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
