# Peer review of "A new and very spiny lizard (Gymnophthalmidae: Echinosaura) from the Andes in northwestern Ecuador"

_PeerJ, doi:10.7717/peerj.12523_

## Round 0.1 · original submission · Minor Revisions

Thanks for this great submission. Essentially, this manuscript is acceptable right now and I'm tempted to say 'accept'. However, Tiffany has made some solid recommendations, and there are some useful annotations from Marco. These are really very minor and I don't think you'll have any difficulty in addressing them.

A minor comment is that you don't need to reduce the number of tables, but we need full legends explaining any abbreviations used.

Reviewer 1 ·

Basic reporting

No comment.

Experimental design

No comment.

Validity of the findings

Evidence presented strongly support conclusion that they have discovered a new species. The authors go far beyond what generally constitutes a species description.

·

Basic reporting

no comment

Experimental design

no comment

Validity of the findings

no comment

Additional comments

I suggest to to add a discussion about the hemipenial morphology of the new species.

·

Basic reporting

The manuscript is clearly written with good English throughout (minor exceptions below). The literature references are sufficient—there is a small body of literature on these taxa and they have included all that was necessary. The article structure was good and clear. They have some excellent useful figures. There are a lot of figures but I believe they are all useful. The ones comparing among the species will be incredibly useful to readers. The skeletal figures are a step above what has ever been done in the past with these taxa. I think there are too many tables and some of the tables could use some work (more below). The paper is self-contained with results relevant to their project.
1. L89-99 and elsewhere in the Methods. Change all of this to active voice to make it easier to read.
2. L155, 158. Poor grammar: “was ran” is incorrect.
3. L231. Tables are referred out of order. You refer to Table 3 here but Tables 1 and 2 later.
4. Table 1. Write out character names. The abbreviations are not defined in this table.
5. Table 2. I’m not sure this table is necessary. They are mostly the same and you can describe the variation in the text. I would omit this table.
6. Table 3. Put the columns in a logical order. I would put the new species first and the rest in alphabetical order (or some other order that makes sense).

Experimental design

This paper contains original research describing a novel species. More importantly for most readers, it clarifies differences among Echinosaura species with helpful tables and figures. It gives excellent skeletal information that is lacking for most gymnophthalmids. The methods are described in sufficient detail to replicate the morphological and molecular analyses.

Validity of the findings

The data appear to be robust and evidence that the species is novel is abundant. The Discussion and Conclusions are quite brief, but I do not think that more needs to be said. I have two comments for the authors to consider that need to be clarified in the text.
1. L177-180. You give ND4 distances for the taxa closely related to your new species but 16S distances for the distantly related ones. Why? Give 16S distances for all of them so that they are comparable in this paper and across other papers.
2. Figure 4. Did you take these photos immediately after they were preserved? If some were preserved longer than others then the patterns may have faded (Doan and Adams, 2015 Zootaxa). The live photos of Figure 6 show the variation but you still have to be careful when displaying patterns of preserved specimens. You should state how long these had been preserved and discuss if there is variation in preservation times.

Additional comments

I look forward to seeing this in print. I would be happy to discuss any of my comments with the authors.

---

## Round 0.2 · accepted · Accept

Thanks for your reworked manuscript. This version is acceptable for publication.